# Sparse Decomposition of Graph Neural Networks

**Yaochen Hu**                                                   *yaochen.hu@huawei.com*
*Huawei Noah's Ark Lab, Montreal, Canada*

**Mai Zeng**                                                       *mai.zeng@mail.mcgill.ca*
*McGill University & Mila & ILLS\*, Montreal, Canada*

**Ge Zhang**                                                      *ge.zhang1@huawei.com*
*Huawei Noah's Ark Lab, Toronto, Canada*

**Pavel Rumiantsev**                                        *pavel.rumiantsev@mail.mcgill.ca*
*McGill University & Mila & ILLS, Montreal, Canada*

**Liheng Ma**                                                     *liheng.ma@mail.mcgill.ca*
*McGill University & Mila & ILLS, Montreal, Canada*

**Yingxue Zhang**                                             *yingxue.zhang@huawei.com*
*Huawei Noah's Ark Lab, Toronto, Canada*

**Mark Coates**                                                  *mark.coates@mcgill.ca*
*McGill University & Mila & ILLS, Montreal, Canada*

**Reviewed on OpenReview:** *https://openreview.net/forum?id=xdWP1d8BxI*

## Abstract

Graph Neural Networks (GNN) exhibit superior performance in graph representation learning, but their inference cost can be high due to an aggregation operation that can require a memory fetch for a very large number of nodes. This inference cost is the major obstacle to deploying GNN models with *online prediction* to reflect the potentially dynamic node features. To address this, we propose an approach to reduce the number of nodes that are included during aggregation. We achieve this through a sparse decomposition, learning to approximate node representations using a weighted sum of linearly transformed features of a carefully selected subset of nodes within the extended neighbourhood. The approach achieves linear complexity with respect to the average node degree and the number of layers in the graph neural network. We introduce an algorithm to compute the optimal parameters for the sparse decomposition, ensuring an accurate approximation of the original GNN model, and present effective strategies to reduce the training time and improve the learning process. We demonstrate via extensive experiments that our method outperforms other baselines designed for inference speedup, achieving significant accuracy gains with comparable inference times for both node classification and spatio-temporal forecasting tasks.

## 1 Introduction

Graph neural networks (GNN) have demonstrated impressive performance for graph representation learning (Hamilton et al., 2017; Veličković et al., 2018; Qu et al., 2019; Rampášek et al., 2022). Although there are numerous designs for GNN models, the essential idea is to represent each node based on its features and its neighbourhood (Wu et al., 2020; Zhou et al., 2020). The procedure of aggregating features from neighbour nodes is empirically and theoretically effective (Xu et al., 2019) in representing the graph structures and

---

\*Mila - Quebec AI Institute and ILLS - International Laboratory on Learning Systems.

blending the features of the nodes. However, deploying GNN models to process large graphs is challenging since collecting information from the neighbour nodes and computing the aggregation is extremely time-consuming (Zhang et al., 2021; Tian et al., 2023; Wu et al., 2023; Liu et al., 2024).

In this work, we tackle the efficient inference problem for GNN models in the *online prediction* setting (Crankshaw, 2019). Specifically, we need to compute the representations of a few arbitrary nodes. The main advantage is that the prediction can reflect potential dynamic features[1] of the input. The computational complexity is dominated by the number of receptive nodes, which rapidly increases as the number of layers in the model grows, for most message-passing-based and graph-transformer-based GNNs (Zeng et al., 2020; Min et al., 2022).

Our goal is to reduce the inference time to linear complexity with respect to the number of layers and the average node degree. Recently, several studies have attempted to address this problem by combining the performance of GNN and the efficiency of MLPs (Zhang et al., 2021; Hu et al., 2021; Tian et al., 2023; Wang et al., 2023; Wu et al., 2023; Liu et al., 2024; Tian et al., 2024; Winter et al., 2024; Wu et al., 2024). Knowledge distillation (Hinton et al., 2015) and feature/label smoothing are used to construct effective MLP models to eliminate the cumbersome neighbour collection and aggregation procedure. Although efficient, these methods have a fundamental limitation: the features gathered at each node are assumed to contain sufficient information to predict the node label accurately. However, to achieve their full potential, especially when features can change at inference time, GNN models should take into account the features from neighbourhood nodes and the graph structure (Battaglia et al., 2018; Pei et al., 2020). Therefore, we ask the question: *given any graph neural network model that relies on both the graph structure and the features of the neighbourhood, can we infer the representation of a node in linear time?*

**Present work.** We propose sparse decomposition for graph neural networks (SDGNN), an approximation to any target GNN models that can infer node representations efficiently and effectively. The SDGNN consists of a feature transformation function and sparse weight vectors for nodes in the graph. The representation of each node is then a weighted sum of the transformed features from a small set of receptive nodes. The sparsity of the weight vectors guarantees low inference complexity. The learnable feature transformation function and the sparse weight vectors grant the SDGNN flexibility to approximate a wide range of targeted GNN models. To find the optimal parameters in SDGNN, we formulate the approximation task as an optimization problem and propose a scalable and efficient solution that iterates between the learning of the transformation function and the optimization of the sparse weight vectors. We verify the approximation power of SDGNN and the scalability of our algorithm on seven node classification datasets and demonstrate how SDGNN can be effectively applied under the online prediction setting with two spatio-temporal forecasting datasets. SDGNN consistently outperforms recent state-of-the-art models designed for GNN inference speedup.

## 2 Preliminaries and Problem Definition

Consider a graph $\mathcal{G} = \{\mathcal{V}, \mathcal{E}\}$ with $|\mathcal{V}| = n$ nodes and $|\mathcal{E}| = m$ edges. Denote by $\mathbf{X} \in \mathbb{R}^{n \times D}$ the feature matrix, with a dimension of $D$, in which the $i^{\text{th}}$ row $\mathrm{X}_{i*}$ denotes the feature vector of node $i$. We address the task of graph representation learning, which involves learning an embedding $g(z, \mathbf{X}|\mathcal{G}) \in \mathbb{R}^d$ for each node $z$, where $d$ denotes the embedding dimension of a GNN. Such representations are fed to additional MLPs $f(\cdot)$ for application in downstream tasks, e.g., node classification/regression, link prediction/regression, and graph classification/regression.

### 2.1 Graph Representation via GNN

We summarize the framework from (Hamilton et al., 2017) as an example to demonstrate the basic concepts of GNNs. We stress that our approach can be applied to any GNN formulation that produces fixed-dimension node embeddings. Let $\boldsymbol{h}_z^l$ be the embedding vector of node $z$ at layer $l$. Denote by $\mathcal{N}(z)$ the set of neighbour

---

[1]The features of a sample may vary over time, e.g., dynamical features from sensors (Dawson et al., 2016), dynamic features in the recommendation systems (Chu & Park, 2009).

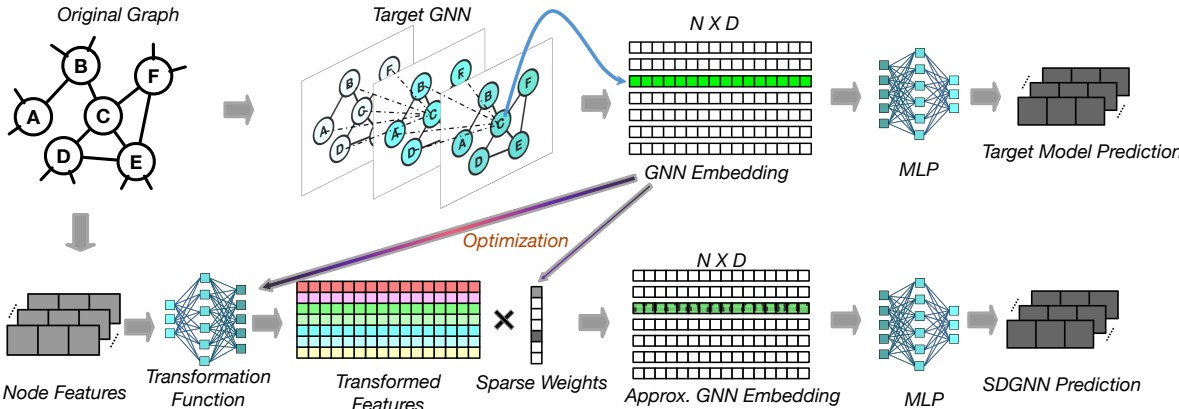

Figure 1: The pipeline overview for SDGNN framework (bottom pipeline). To compute GNN embedding efficiently, we use a transformation function to adapt node features and introduce sparse vectors associated with each node to gather information from critical neighbours. The parameters in the transformation function and the sparse vectors are determined by optimization to approximate the target GNN embeddings.

nodes of $z$. $\boldsymbol{h}_z^l$ is iteratively evaluated as

$$\boldsymbol{h}_{\mathcal{N}(z)}^l = \text{AGGREGATE}_l\left(\{\boldsymbol{h}_v^{l-1}|\forall v \in \mathcal{N}(z)\}\right),$$
$$\boldsymbol{h}_z^l = \text{UPDATE}_l(\boldsymbol{h}_z^{l-1}, \boldsymbol{h}_{\mathcal{N}(z)}^l),$$

where $\text{AGGREGATE}_l(\cdot)$ is any *permutation invariant* operation that aggregates a set of vectors, and $\text{UPDATE}_l(\cdot)$ is any transformation function, possibly incorporating learnable parameters and non-linear transformations. $\boldsymbol{h}_z^0$ is usually initialized as the node features of $z$, i.e., $\mathrm{X}_{z*}$.

$$g(z, \mathbf{X}|\mathcal{G}) := \boldsymbol{h}_z^L, \tag{1}$$

where $L$ is a predefined hyperparameter indicating the number of layers. An additional decoder module $f(\cdot)$ can be applied over $g(z, \mathbf{X}|\mathcal{G})$ to map the GNN representation to the target label $f(g(z, \mathbf{X}|\mathcal{G}))$.

## 2.2 Online Prediction for Node Representations from GNN

*Online predicton* is a model serving paradigm that computes the prediction of a few arbitrary samples upon request (Crankshaw, 2019). It has stringent requirements for low latency and high throughput. The advantage of online prediction is that it reflects dynamic features. In the case of node-level online prediction for GNNs, we need to compute the embeddings in (1) of a few nodes efficiently upon request.

The complexity of computing a node representation in (1) is proportional to the number of receptive nodes. Specifically, for the message-passing-based models, the number of receptive nodes potentially grows exponentially with the number of GNN layers (Zeng et al., 2020). For a graph transformer, one of the primary motivations is to enlarge the number of receptive nodes even if they are not connected to the target node, so the number of receptive nodes is usually larger than the message-passing-based models (Min et al., 2022). Therefore, deploying an online prediction system for GNN models is challenging due to the potentially very large number of receptive nodes. For any GNN model, we aim to find an approximate computation $\hat{g}(z, \mathbf{X}|\mathcal{G})$, so that the *inference complexity* is linear with respect to the number of GNN layers $L$ and the average node degree $\bar{d}$, i.e., $O(\bar{d}L)$. We require that $\hat{g}(z, \mathbf{X}|\mathcal{G})$ should be close to $g(z, \mathbf{X}|\mathcal{G})$ for $\forall z \in \mathcal{V}$.

**Remarks.** We aim to invent a fundamental solution for the challenging case where prediction could benefit from potentially dynamic node features, especially for those from a few critical neighbours. For example, the recent purchasing behaviour of one's closest friend might probably influence her online shopping preference; the traffic flow of an intersection in the near future should be correlated with the status of its neighbouring intersections, etc. Therefore, we want to develop a technique to efficiently summarize or approximate the

Table 1: Comparison of the related techniques. *Inference complexity* depicts the asymptotic complexity measured by the number of receptive nodes. *Memory overhead is the overall memory required. Neighbour feature-aware* indicates whether the inference model can reflect neighbour features. *Dynamic Features* represents whether the inference model can be efficiently applied with dynamic features. $L$: number of the target GNN layers; $L'$: number of a shallower GNN layers where $L' < L$; $|\mathcal{V}|$: node number; $\bar{d}$: average node degree; $s$: sampling budget; $D$: node feature dimension.

| Related Techniques | Inference complexity | Memory overhead | Neighbour feature-aware | Dynamic features |
|---|---|---|---|---|
| Neighbour Sampling | $O(s^L)$ | $O(|\mathcal{V}|\bar{d} + |\mathcal{V}|D)$ | ✔ | ✔ |
| Embedding Compression | $O(\bar{d}^L)$ | $O(|\mathcal{V}|\bar{d} + |\mathcal{V}|D)$ | ✔ | ✔ |
| GNN Knowledge Distillation | $O(\bar{d}^{L'})$ | $O(|\mathcal{V}|\bar{d} + |\mathcal{V}|D)$ | ✔ | ✔ |
| MLP Knowledge Distillation | $O(1)$ | $O(|\mathcal{V}|D)$ | ✘ | ✔ |
| Heuristic Decoupling | $O(1)$ | $O(|\mathcal{V}|D)$ | ✔ | ✘ |
| SDGNN (ours) | $O(\bar{d}L)$ | $O(|\mathcal{V}|\bar{d}L + |\mathcal{V}|D)$ | ✔ | ✔ |

dynamic and potentially critical neighbour features to fully convey the power of GNNs. Without dynamic node features, we always have an easy workaround method to cache all the predictions instead of cumbersome computation upon request; without the critical dependency on neighbour features, one could train a decent model without GNN to get an optimized performance.

## 3 Related Work

To the best of our knowledge, we are the first work to study the efficient inference of GNN models under the challenging but rewarding online prediction setting. Some simple adaptions of existing works can partially solve this problem but cannot tackle the intrinsic challenges.

**Neighbourhood Sampling** methods process subsets of edges for each node, selected according to statistical approaches. Hamilton et al. (2017) propose recursively sampling a fixed number of neighbours. However, this approach still suffers from an exponentially growing aggregation complexity. The layer-wise and graph-wise sampling techniques (Chen et al., 2018; Zou et al., 2019; Chiang et al., 2019; Zeng et al., 2020) enhance node-wise sampling by using the common neighbours of nodes in the same mini-batch, but they are optimized for training, and they are not suitable for the online prediction setting.

**Embedding Compression** methods reduce inference time by compressing embeddings with lower accuracy. Zhao et al. (2020) use neural architecture search over the precision of parameters and adopts attention and pooling to speed up inference. Zhou et al. (2021) compress the embedding dimension by learning a linear projection. Ding et al. (2021) employ vector quantization to approximate the embeddings of subsets of nodes during training. Tan et al. (2020) hash the final representations. These works do not reduce the complexity of neighbour fetching and aggregation.

**GNN Knowledge Distillation** techniques employ GNN models as teachers to train students of smaller GNN models for inference. Gao et al. (2022) distill the original GNN to a GNN with fewer layers. Yan et al. (2020) learn one GNN layer with self-attention to approximate each 2 GNN layers, which halves the total number of GCN layers in the student model. These methods trade off some performance for better efficiency. However, the student is still a GNN model which suffers from the large number of receptive nodes.

**MLP Knowledge Distillation.** Recently, a series of works dramatically reduced the inference complexity by training MLP-based models solely on the (augmented) features of the target node to approximate GNN representations (Yang et al., 2024; Zhang et al., 2021; Hu et al., 2021; Tian et al., 2023; Wang et al., 2023; Wu et al., 2023; Liu et al., 2024; Tian et al., 2024). Moreover, Winter et al. (2024); Wu et al. (2024) directly learn MLP models with label smoothing regularized by the graph structure instead of learning from a trained GNN model. However, these models cannot consider the features of neighbour nodes during inference and are prone to have limited expressive power compared to general GNN models.

**Heuristic Decoupling.** Bojchevski et al. (2020) use personalized PageRank to construct shallow GNN weights instead of performing layer-wise message passing. Duan et al. (2022); Chen et al. (2021); Wu et al. (2019); Chen et al. (2019); Nt & Maehara (2019); Rossi et al. (2020); He et al. (2020) generate augmented, informative node features using techniques such as message passing or PageRank. Winter et al. (2024) pre-compute propagated labels from the training set and add them to the node features. During inference, the models rely solely on these cached augmented node features. These hand-crafted rules to generate informative node features are prone to be sub-optimal. Besides, under the online prediction setting, heuristic decoupling methods have to periodically update the augmented features to support dynamic features, which have great computational overhead.

**Position of Our Work:** Compared to existing works, our proposed SDGNN offers linear complexity in terms of inference time but can still process the most relevant neighbour features during inference. Table 1 depicts a detailed illustration of our work's position with asymptotic analysis on memory overhead and inference complexity.

# 4 Sparse Decomposition of Graph Neural Networks

We aim to develop a flexible approach to approximate the node representation from a wide range of GNN models but maintain low *inference complexity*. To this end, we introduce a feature transformation function $\phi(\cdot; \mathbf{W}) : \mathbb{R}^D \to \mathbb{R}^d$ that maps each row of $\mathbf{X} \in \mathbb{R}^{|\mathcal{V}| \times D}$ to dimension of the target GNN representation $d$, where $\mathbf{W}$ represents the learnable parameters. For simplicity, we extend the $\phi$ notation to the matrix input $\mathbf{X}$ to $\phi(\mathbf{X}; \mathbf{W}) \in \mathbb{R}^{|\mathcal{V}| \times d}$ where $\phi(\cdot; \mathbf{W})$ is applied to each row of $\mathbf{X}$. For each node $z$, we define a sparse vector $\boldsymbol{\theta}_z \in \mathbb{R}^{|\mathcal{V}|}$ and model the representation of node $z$ as a linear combination over the transformed node features via $\boldsymbol{\theta}_z$. We thus define the *sparse decomposition of graph neural network* for node $z$ as

$$\hat{g}(z, \mathbf{X}|\mathcal{G}) := \boldsymbol{\theta}_z^{\mathsf{T}} \phi(\mathbf{X}; \mathbf{W}). \tag{2}$$

Intuitively, the sparsity of $\boldsymbol{\theta}_z$ and the node-wise dependency on $\phi(\cdot; \mathbf{W})$ ensure that the computation of $\hat{g}(z, \mathbf{X}|\mathcal{G})$ depends on a limited set of node features. Hence, the inference complexity can be controlled to be $O(dL)$. The learnable $\boldsymbol{\theta}_z$ and $\phi(\cdot; \mathbf{W})$ give SDGNN sufficient flexibility to approximate a wide range of GNN models.

**Matrix form.** Let $(\boldsymbol{\theta}_1, \boldsymbol{\theta}_2, \ldots, \boldsymbol{\theta}_{|\mathcal{V}|})$ be the columns of the matrix $\boldsymbol{\Theta}$. Denote the node representation matrix as $\boldsymbol{\Omega} \in \mathbb{R}^{|\mathcal{V}| \times d}$, where the $z^{th}$ row $\Omega_{z*} := g(z, \mathbf{X}|\mathcal{G})$. Correspondingly, we define $\hat{\boldsymbol{\Omega}} := \boldsymbol{\Theta}^{\mathsf{T}} \phi(\mathbf{X}; \mathbf{W})$.

## 4.1 Relation to Existing Methods

We compare SDGNN to the types that have linear complexity summarized in Table 1. If we assign an identity matrix to $\boldsymbol{\Theta}$, the SDGNN degrades to a model solely based on the self-node features. This covers most of the methods under the umbrella of MLP knowledge distillation. For models that augment the node features, like NOSMOG Tian et al. (2023), SDGNN can also trivially adopt such features. Moreover, if we fix $\boldsymbol{\Theta}$ as the personalized PageRank matrix or power of the normalized adjacent matrix, SDGNN is equivalent to the heuristic decoupling approach. Therefore, SDGNN has at least the expressive power of the models from MLP knowledge distillation and heuristic decoupling.

# 5 SDGNN Computation

We formulate the task of identifying the optimal $\boldsymbol{\Theta}$ and $\phi(\cdot; \mathbf{W})$ as an optimization problem:

$$\underset{\boldsymbol{\Theta}, \phi(\cdot; \mathbf{W})}{\text{minimize}} \quad \mathcal{L}(\boldsymbol{\Theta}, \mathbf{W}) = \frac{1}{2}\|\boldsymbol{\Theta}^{\mathsf{T}}\phi(\mathbf{X}; \mathbf{W}) - \boldsymbol{\Omega}\|_F^2 + \lambda_1\|\boldsymbol{\Theta}\|_{1,1} + \lambda_2\|\mathbf{W}\|_F^2, \tag{3}$$

where $\lambda_1 \geq 0$ is a hyperparameter that controls the sparsity of $\boldsymbol{\Theta}$ via the column-wise L1 regularization term $\|\boldsymbol{\Theta}\|_{1,1}$, and $\lambda_2 \geq 0$ is the hyperparameter for L2 regularization of $\mathbf{W}$. In common models such as a multi-layer perceptron (MLP), the L2 regularization implicitly upper bounds the row-wise norm of $\phi(\mathbf{X}; \mathbf{W})$ for a given $\mathbf{X}$. This prevents the degenerate case of an extremely sparse $\boldsymbol{\Theta}$ with small elements.

## 5.1 Optimization

Jointly learning $\boldsymbol{\Theta}$ and $\mathbf{W}$ is challenging due to the sparsity constraint on $\boldsymbol{\Theta}$. We optimize them iteratively, fixing one while updating the other, termed Phase $\boldsymbol{\Theta}$ and Phase $\phi$.

**Phase $\boldsymbol{\Theta}$.** For each node $z \in \mathcal{V}$, we update $\boldsymbol{\theta}_z$ with the solution of the following optimization problem:

$$\underset{\boldsymbol{\theta}}{\operatorname{argmin}} \quad \frac{1}{2}\|\boldsymbol{\theta}^{\intercal}\phi(\mathbf{X};\mathbf{W}) - \boldsymbol{\Omega}_{z*}\|_2^2 + \lambda_1\|\boldsymbol{\theta}\|_1 \tag{4}$$

$$\text{s.t.} \quad \boldsymbol{\theta} \geq 0. \tag{5}$$

The constraints in (5) are not necessary for the general case, but we empirically find that it makes the optimization procedure more robust. We adopt Least Angle Regression (LARS) (Efron et al., 2004) to solve this Lasso problem, where the maximum number of receptive nodes can be controlled explicitly by setting up the maximum iteration of LARS.

**Phase $\phi$.** We update $\mathbf{W}$ with the solution of the following optimization problem:

$$\underset{\mathbf{W}}{\operatorname{argmin}} \quad \frac{1}{2}\|\boldsymbol{\Theta}^{\intercal}\phi(\mathbf{X};\mathbf{W}) - \boldsymbol{\Omega}\|_F^2 + \lambda_2\|\mathbf{W}\|_F^2, \tag{6}$$

and solve it with the gradient descent (GD) algorithm. For efficient training, we take the $\mathbf{W}$ from its last iteration as a warm start and only update it for a few steps instead of reaching a converged solution.

We iteratively update $\boldsymbol{\Theta}$ and $\mathbf{W}$ until the change of the loss $\mathcal{L}$ in (3) is smaller than some predefined threshold.

**Convergence.** Let $\boldsymbol{\Theta}_t, \mathbf{W}_t$ be the parameters after the $t$th iteration with the corresponding loss $\mathcal{L}(\boldsymbol{\Theta}_t, \mathbf{W}_t)$. From the definition of loss in (3), we have $\mathcal{L}(\boldsymbol{\Theta}_t, \mathbf{W}_t) \geq 0$ for $\forall t$. From the alternative optimization procedure with a proper setting of learning rate for Phase $\mathbf{W}$, for $\forall t$, we also have

$$\mathcal{L}(\boldsymbol{\Theta}_t, \mathbf{W}_t) \geq \mathcal{L}(\boldsymbol{\Theta}_{t+1}, \mathbf{W}_t) \geq \mathcal{L}(\boldsymbol{\Theta}_{t+1}, \mathbf{W}_{t+1}).$$

Therefore, $\mathcal{L}(\boldsymbol{\Theta}_t, \mathbf{W}_t)$ is guaranteed to converge as $t$ grows.

**Per iteration complexity.** Since $\mathcal{L}$ is non-convex, providing the asymptotic convergence rate is challenging. We analyze the per-iteration complexity to get insight into the empirical computation overhead. For Phase $\boldsymbol{\Theta}$, we first need to compute $\phi(\mathbf{X};\mathbf{W})$, which has a complexity of $O(Dd + L'd^2)$ for each node, where $L'$ is the number of layers in the MLP $\phi$ and assuming the hidden dimension is always $d$. The complexity of computing the solution for each node is $O(|\mathcal{V}|^3 + |\mathcal{V}|^2 d)$ (Efron et al., 2004)[2]. Then the complexity for Phase $\boldsymbol{\Theta}$ is $O(|\mathcal{V}|(Dd + L'd^2 + |\mathcal{V}|^3 + |\mathcal{V}|^2 d))$. For Phase $\mathbf{W}$, assuming we need $k$ gradient descent iterations, the complexity is $O(k|\mathcal{V}|(Dd + L'd^2))$. Overall, the per-iteration complexity is $\boldsymbol{\Theta}$ is $O(|\mathcal{V}|(k(Dd + L'd^2) + |\mathcal{V}|^3 + |\mathcal{V}|^2 d))$, and and the asymptotic complexity for $|\mathcal{V}|$ is $O(|\mathcal{V}|^4)$. The $O(|\mathcal{V}|^4)$ factor is the bottleneck for efficient training for large graphs.

## 5.2 Scaling to Large Graphs

The naive optimization algorithm does not scale to large graphs due to the bottleneck at Phase $\boldsymbol{\Theta}$, which has an asymptotic per-iteration complexity of $O(|\mathcal{V}|^4)$. We propose two main techniques to tackle the scalability issue: stochastic mini-batch training and narrowing the candidate nodes.

**Stochastic mini-batch Training.** We perform stochastic mini-batch updates. Specifically, we randomly select $\mathbb{B} \subset \mathcal{V}$ nodes at each iteration and only update the $\boldsymbol{\theta}_z$ where $z \in \mathbb{B}$. As for $\mathbf{W}$, we select the same subset $\mathbb{B}$ of nodes and the corresponding rows of $\boldsymbol{\Theta}^{\intercal}\phi(\mathbf{X};\mathbf{W}) - \boldsymbol{\Omega}$ to include in the loss term for each mini-batch, i.e., we solve:

$$\underset{\mathbf{W}}{\operatorname{argmin}} \quad \frac{1}{2}\sum_{z \in \mathbb{B}}\|\boldsymbol{\theta}_z^{\intercal}\phi(\mathbf{X};\mathbf{W}) - \boldsymbol{\Omega}_{z*}\|_2^2 + \lambda_2\|\mathbf{W}\|_F^2. \tag{7}$$

---

[2]Efron et al. (2004) mentioned that the complexity is $O(d^3)$ with saturated least-squares fit when $|\mathcal{V}| \gg d$. We adopted the implementation from the scikit-learn library, and empirically, it's always $O(|\mathcal{V}|^3 + |\mathcal{V}|^2 d)$. From the practical point of view, we count the complexity to always be $O(|\mathcal{V}|^3 + |\mathcal{V}|^2 d)$

---

**Algorithm 1** SDGNN Computation

---

 1: **procedure** SDGNN($\mathbf{\Omega}, \mathcal{G}, \mathbf{X}$)
 2:     Prepare the candidate sets $\mathbb{C}_z$ for $\forall z \in \mathcal{V}$ according to Narrowing the Candidate Nodes.
 3:     **for** $t = 1$ to $T$ **do**                                                                     ▷ Main training loop.
 4:         Get the current mini-batch of the node set $\mathbb{B}_t \subset \mathcal{V}$.
 5:         **for** each $z$ in $\mathbb{B}_t$ **do**
 6:             Update $\boldsymbol{\theta}_z$ with Phase $\mathbf{\Theta}$.
 7:         **end for**
 8:         Update $\mathbf{W}$ with Phase $\phi$.
 9:     **end for**
10:     **for** $z$ in $\mathcal{V}$ **do**                                                                ▷ Fix $\mathbf{W}$ and finalize the $\mathbf{\Theta}$.
11:         Update $\boldsymbol{\theta}_z$ with Phase $\mathbf{\Theta}$.
12:     **end for**
13:     **return** $\mathbf{W}$ and $\theta_z$ for $\forall z \in \mathcal{V}$.
14: **end procedure**

---

**Narrowing the candidate nodes.** The computation required to solve (4) rapidly grows as the size of $\mathcal{V}$ increases. Specifically, the computational overhead comes from the inference of $\phi(\mathbf{X}; \mathbf{W})$ and the computation time of the LARS solver. To reduce the required computation, we define a much smaller candidate node set $\mathbb{C}_z$ for each node $z$ and only consider the candidate nodes to determine $\boldsymbol{\theta}_z$. We solve the following problem instead of (4).

$$\underset{\boldsymbol{\theta}}{\arg\min} \quad \frac{1}{2}\|\boldsymbol{\theta}^{\mathsf{T}}\phi(\mathbf{X}; \mathbf{W}) - \mathbf{\Omega}_{z*}\|_2^2 + \lambda_1\|\boldsymbol{\theta}\|_1 \tag{8}$$

$$\text{s.t.} \quad \boldsymbol{\theta}_i \geq 0 \quad \forall i \in \mathbb{C}_z, \quad \boldsymbol{\theta}_i = 0 \quad \forall i \notin \mathbb{C}_z. \tag{9}$$

(9) defines the reduced candidate set for solving $\theta_z$. We only need to infer $\phi(\mathbf{X}_{z*}; \mathbf{W})$ for node $z$ that appears in the candidate set, and the complexity of the LARS solver only depends on the size of the candidate set instead of $|\mathcal{V}|$.

For each node, we use the knowledge of the graph structure to heuristically determine the moderate-sized candidate set that includes the most relevant nodes. Specifically, we include all $K_1$-hop neighbour nodes. From those nodes, we recursively sample a fixed number $s$ of neighbours for an extra $K_2$ hops (similar to the mechanism of graphSAGE (Hamilton et al., 2017)) and combine all visited nodes as the candidate set. As for practical guidelines for selecting $K_1$, $K_2$ and $s$, we can refer to the receptive nodes of the target GNN models and set the hyper-parameters so that the selected candidate nodes can roughly cover them. After that, a grid search could be conducted around that set of hyper-parameters to get better performance.

**Per iteration complexity.** The size of $\mathbb{C}_z$ is roughly $O(\bar{d}^{K_1}s^{K_2})$, where $\bar{d}$ is the average node degree and $s$ is the sampling budget, and $|\mathbb{C}_z| \ll |\mathcal{V}|$ for very large graphs. Then in Phase $\mathbf{\Theta}$, the complexity for each node is $O(|\mathbb{C}_z|^3 + |\mathbb{C}_z|^2 d)$. With similar analysis in Sec. 5.1, the overall complexity will be $O(|\mathbb{B}|(k(Dd + L'd^2)|\mathbb{C}_z| + |\mathbb{C}_z|^3 + |\mathbb{C}_z|^2 d))$, and the asymptotic complexity is $O(|\mathbb{B}||\mathbb{C}_z|^3)$. We can see that the asymptotic per-iteration complexity is independent of the size of the graph $|\mathcal{V}|$.

Considering the design strategies outlined above, the practical algorithm is presented in Algorithm 1. We add an extra loop at Line 10 to align all $\boldsymbol{\theta}$ with the latest version of $\mathbf{W}$.

# 6 Experiments

## 6.1 Node Classification Tasks

We validate SDGNN's approximation power and inference efficiency through node classification tasks in the transductive setting.

### 6.1.1 Task

We consider the node classification task under the *transductive* setting. Given a graph $\mathcal{G} = \{\mathcal{V}, \mathcal{E}\}$ with $|\mathcal{V}| = n$ nodes and $|\mathcal{E}| = m$ edges, the feature matrix of nodes $\mathbf{X} \in \mathbb{R}^{n \times D}$ and the labels from a subset of nodes $\mathcal{V}_{\text{train}} \subset \mathcal{V}$, the task is to predict the labels for the remaining nodes.

### 6.1.2 Datasets

Following Zhang et al. (2021) and Tian et al. (2023), we conduct experiments on five widely used benchmark datasets from Shchur et al. (2018), namely, Cora, Citeseer, Pubmed, Computer and Photo. We also examine performance on two large-scale datasets, Arxiv and Products, from the OGB benchmarks (Hu et al., 2020).

### 6.1.3 Baselines and Target GNN Models

We select GLNN (Zhang et al., 2021), NOSMOG (Tian et al., 2023), CoHOp (Winter et al., 2024), SGC (Wu et al., 2019), and PPRGo (Bojchevski et al., 2020) as the baselines. Specifically, GLNN and NOSMOG distill the GNN into an MLP with and without augmenting with position encodings. CoHOp is a recent method that trains an MLP to replace the GNN, using label propagation to incorporate graph information. SGC and PPRGo adopt the heuristic of selecting a critical subset of nodes and assigning aggregation weights. To make a fair comparison, we match the number of receptive nodes to SDGNN by truncating, where for each node $z$, we keep the top $k_z$ most significant weights, $k_z$ being the corresponding number of receptive nodes in SDGNN at node $z$. We also include a degraded version of SDGNN where for each $\theta_z$, we replace the learned non-zero weights with normalized equal weights (SDGNN-Equal).

Regarding the targeted GNN models, we include GraphSAGE (Hamilton et al., 2017) with mean aggregator. This is the base model for all experiments reported in Zhang et al. (2021) and Tian et al. (2023). To fully evaluate the capacity of each efficient inference solution, we also add one of the most effective GNN models for each dataset. In this selection process, we exclude the models that achieve impressive performance via processing steps unrelated to graph learning, e.g., label re-use (Wang & Leskovec, 2021) and feature augmentation. Given that no single GNN model achieves the best performance across all datasets, we select distinct target models that excel on each dataset. Specifically, we choose Geom-GCN (Pei et al., 2020) for Cora, Citeseer and Pubmed. We adopt Exphormer (Shirzad et al., 2023) for Computer and Photo. We select DRGAT (Zhang et al., 2023) for Arxiv and RevGNN-112 (denoted by RevGNN) (Li et al., 2021) for Products.

### 6.1.4 Evaluation

For the 5 small datasets (Cora, Citeseer, Pubmed, Computer and Photo), we randomly split the nodes with a 6:2:2 ratio into training, validation and testing sets. Experiments are conducted using 10 random seeds, as in Pei et al. (2020). We report the mean and standard deviation. For Arxiv and Products, we follow the fixed predefined data splits specified in Hu et al. (2020), run the experiments 10 times, and report the mean and standard deviation.

### 6.1.5 SDGNN Integration

We start with a trained GNN model, e.g., GraphSAGE (Hamilton et al., 2017), DeeperGCN (Li et al., 2021), or graph transformers (Yun et al., 2019; Rampášek et al., 2022; Ma et al., 2023; Shirzad et al., 2023). We adopt the intermediate representation constructed by the architecture immediately before the final prediction as the target node representation $g(z, \mathbf{X}|\mathcal{G})$, and learn the SDGNN $\hat{g}(z, \mathbf{X}|\mathcal{G})$. Using the label and the representation $\hat{g}(z, \mathbf{X}|\mathcal{G})$ for the nodes within $\mathcal{V}_{\text{train}}$, we train a decoder function $f(\hat{g}(z, \mathbf{X}|\mathcal{G}))$ to map the node representation to the predicted labels. We employ an MLP for the decoder model. During inference, upon a request for prediction for a node $z$, we refer to $\boldsymbol{\theta}_z$ to retrieve the receptive node features and compute the prediction with $f(\hat{g}(z, \mathbf{X}|\mathcal{G}))$ over the receptive nodes. The overall pipeline of integrating the SDGNN is depicted in Figure 1.

**Remark.** Our method requires the predicted representation from the GNN for *all* nodes to get optimized $\theta_z$. Although the approximation with static node features has limited practical value, we include these

Table 2: Mean F1 micro score and std. dev. for node classification tasks, best in **bold** and second best underlined. * marks statistically significant results under Wilcoxon signed-rank test (Wilcoxon, 1992) at the 5% significance level.

| Data/Target Model | Target Model | GLNN | NOSMOG | CoHOp | PPRGo | SGC | SDGNN-Equal | SDGNN(ours) |
|---|---|---|---|---|---|---|---|---|
| Cora/SAGE | 85.43±1.22 | 86.86±1.39 | 85.82±1.39 | 85.49± 0.89 | 86.41±1.39 | **87.23±1.40** | 84.21±1.53 | 85.53±1.32 |
| Cora/Geom-GCN | 86.52±0.98 | 85.78±1.26 | 85.68±1.01 | 85.49± 0.89 | 85.40±1.37 | 85.99±1.38 | 80.79±1.31 | **86.64±0.85** |
| Citeseer/SAGE | 72.40±1.79 | 76.89±1.53 | 76.41±1.39 | 73.22 ± 1.21 | **77.19±1.16** | 76.29±1.37 | 73.13±1.95 | 73.62±2.04 |
| Citeseer/Geom-GCN | 79.91±0.94 | 79.34±1.40 | 79.88±0.93 | 73.22 ± 1.21 | 77.37±0.65 | 76.83±0.82 | 76.63±1.11 | **80.27±1.11** |
| Pubmed/SAGE | 87.17±0.56 | 89.21±0.66 | 87.98±0.62 | 84.81± 0.56 | **89.40±0.45** | 86.77±0.52 | 85.42±0.71 | 87.10±0.54 |
| Pubmed/Geom-GCN | 89.75±0.43 | **90.88±0.53** | 90.31±0.59 | 84.81± 0.56 | 89.35±0.38 | 86.24±0.76 | 85.81±0.46 | 89.77±0.47 |
| Computer/SAGE | 89.06±0.53 | 88.49±0.72 | 89.29±0.74 | **91.28±0.47** | 88.98±0.43 | 90.52±0.59 | 89.85±0.56 | 90.60±0.52 |
| Computer/Exphormer | 94.19±0.56 | 92.24±0.71 | 93.28±0.62 | 91.28 ± 0.47 | 89.45±0.60 | 90.71±0.65 | 93.44±0.58 | **94.29±0.52**\* |
| Photo/SAGE | 92.90 ±0.67 | 93.87±0.33 | 94.27±0.50 | **95.48±0.31** | 94.93±0.49 | 93.54±0.55 | 93.01±0.59 | 93.96±0.39 |
| Photo/Exphormer | 96.54±0.35 | 95.58±0.56 | 95.35±0.52 | 95.48 ± 0.31 | 94.95±0.34 | 93.90±0.52 | 96.35±0.32 | **96.73±0.30**\* |
| Arxiv/SAGE | 70.23±0.23 | 63.40±0.23 | 69.78±0.22 | **72.79±0.09** | 66.05±0.24 | 69.95±0.29 | 69.83±0.24 | 70.50±0.33 |
| Arxiv/DRGAT | 73.78±0.09 | 65.92±0.17 | 72.16±0.15 | 72.79±0.09 | 68.16±0.19 | 72.09±0.14 | 73.45±0.10 | **73.72±0.11**\* |
| Products/SAGE | 78.37±0.55 | 62.87±0.53 | 77.33±0.32 | **81.67±0.25** | 70.15±0.59 | 74.97±0.61 | 78.03±0.65 | 78.07±0.61 |
| Products/RevGNN | 82.79±0.25 | 64.49±0.17 | 80.91±0.23 | 81.67±0.25 | 73.93±0.20 | 79.76±0.28 | 82.64±0.22 | **82.87±0.21**\* |

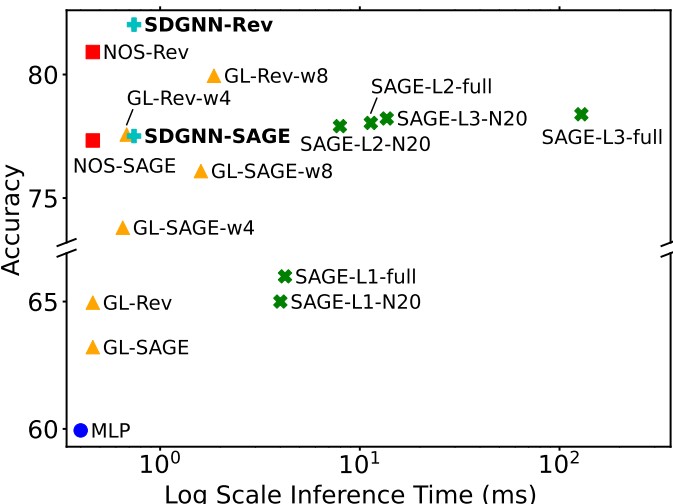

Figure 2: Accuracy v.s. mean inference wall-clock time over 10,000 randomly sampled nodes on the Products test set. GL→GLNN, NOS→NOSMOG, Rev→RevGNN. w4, w8: student size enlarged 4, 8 times. L1, L2, and L3 denote GNN layers. N20: a neighbour sampling size of 20.

experiments to compare the approximation power and efficiency of the proposed SDGNN with the common benchmarks. The primary use-case for SDGNN is when there are dynamic node features under the *online prediction* setting. We demonstrate this via a spatio-temporal forecasting task in later sections.

### 6.1.6 Main Results

We compare the accuracy among GLNN, NOSMOG, CoHOp, PPRGo, SGC and SDGNN for seven datasets with different target GNN models in Table 2. The "Target Model" column indicates the performance of the original GNN models. CoHOp, SGC and PPRGo do not rely on the target GNN embeddings. We duplicate the results of CoHOp within each dataset to compare the performance better. SGC and PPRGo have different performances under different target GNN models since we truncate the neighbours according to the patterns from the corresponding SDGNN models. Overall, SDGNN can approximate the target for all scenarios well, achieving accuracy close to the target model. Other techniques like NOSMOG, CoHOP, etc., though provided with a target GNN model, are not designed to solely approximate the target but also integrate other modules to intervene in the outputs. As a result, with a weaker target GNN model, SDGNN

Table 3: Mean Absolute Percentage Error (MAPE, %) for next-step forecasting on the PeMS04 and PeMS08 datasets. The receptive field size of the SDGNN model is set to 6 for both datasets. * indicates SDGNN is stat. significantly better than the next-best (excluding the target), for a paired Wilcoxon signed-rank test at the 5% significance level.

| MAPE | GRU-GCN | GRU | GLNN | NOSMOG | PPRGo | SDGNN |
|---|---|---|---|---|---|---|
| **PeMS04** | $1.17_{\pm 0.09}$ | $2.09_{\pm 0.6}$* | $1.51_{\pm 0.10}$* | $1.53_{\pm 0.31}$* | $2.63_{\pm 0.18}$* | $\mathbf{1.33}_{\pm 0.09}$ |
| **PeMS08** | $0.89_{\pm 0.03}$ | $1.69_{\pm 0.06}$* | $1.55_{\pm 0.08}$* | $1.68_{\pm 0.03}$* | $1.63_{\pm 0.07}$* | $\mathbf{0.97}_{\pm 0.04}$ |

can be outperformed by those techniques. In contrast, with SOTA GNN targets, SDGNN achieves the best performance in 6/7 scenarios, including 2/2 scenarios on large-scale datasets.

Specifically, GLNN and NOSMOG often outperform the SAGE target model, but the performance gap is more pronounced for GLNN on larger datasets like Arxiv and Products. For example, on Products/RevGNN the performance gap for GLNN is 18.30 percent. An essential difference between GLNN and NOSMOG is that NOSMOG calculates a positional encoding (derived using DeepWalk) for each node and stores this as an additional feature. This is highly effective for the Products and Arxiv datasets. CoHOp augments the node features with neighbour labels and trains an MLP with a label smoothing loss on the graph. It outperforms the weak SAGE models for all datasets, but it cannot achieve accuracy levels close to those of the SOTA target GNN models. PPRGo and SGC adopt different heuristics to select the neighbours, and their performance varies across different datasets. Due to the optimized neighbour selection and weight assignment, SDGNN outperforms these heuristic-based methods for all scenarios involving more powerful target GNN models.

**Ablation study:** Replacing the learned $\theta_z$ with normalized equal values decreases the performance. This demonstrates that SDGNN locates important neighbours and assigns appropriate aggregation weights to approximate target GNN embeddings more effectively. Even with normalized weights, the approximation still outperforms the other baselines for the scenarios with larger graphs and more powerful target GNN models. This highlights that the receptive node selection aspect of SDGNN is highly effective.

### 6.1.7 Inference Time

We performed inference for $10,000$ randomly sampled nodes for each dataset to assess the trade-off between inference time and accuracy. Here, we provide results and a discussion of the Products dataset; the results for other datasets are qualitatively similar and are provided in the Appendix. Fig. 2 illustrates the testing accuracy versus the average computation time per node for different models and approximation techniques. We observe that SDGNN-Rev (SDGNN based on RevGNN) achieves the best accuracy (82.87%) with inference time (0.74 ms), while the MLP-based methods (MLP, GL-Rev, GL-SAGE, NOS-SAGE, NOS-Rev) have the fastest inference time around 0.45 ms. SDGNN is slower due to the additional computation costs of performing feature transformation on multiple nodes and aggregation to incorporate neighbour features.

The SAGE series, such as SAGE-L3-full (a 3-layer SAGE model without sampling during inference) and SAGE-L2-N20 (a 2-layer SAGE model that samples 20 neighbours), exhibit rapidly growing inference times as the number of layers increases (up to 128 ms), making them impractical for real-world applications. We also detail the results for GL-SAGE-w4 (GL-SAGE with hidden dimension 4 times wider) and GL-Rev-w4 (a 4 times wider version of GL-Rev). Although their accuracy improves compared to their base versions, their inference times also increase substantially. We conclude that SDGNN offers superior performance in terms of accuracy, and its inference time is sufficiently close to the fastest MLP models.

### 6.2 Spatio-Temporal Forecasting

We demonstrate how SDGNN can be applied to tasks with dynamic node features through spatio-temporal forecasting tasks, which are aligned with the online prediction setting.

### 6.2.1 Task

We have a graph $\mathcal{G} = \{\mathcal{V}, \mathcal{E}\}$ with $|\mathcal{V}| = n$ nodes and $|\mathcal{E}| = m$ edges. The nodes are associated with signals in discrete time $\mathbf{X}_{1:T} = (\mathbf{X}_1, \mathbf{X}_2, \ldots, \mathbf{X}_T) \in \mathbb{R}^{n \times T \times C}$, and each $\mathbf{X}_t \in \mathbb{R}^{n \times C}$ denotes $C$ channel signals for all $n$ nodes at time $t$. The task is to predict $\mathbf{X}_{T+1} \in \mathbb{R}^{n \times C}$.

### 6.2.2 Datasets

We consider the traffic datasets PeMS04 and PeMS08 from Guo et al. (2021) and apply the data preprocessing of Gao & Ribeiro (2022). Given a fixed graph $\mathcal{G}$, the signals on the nodes are partitioned into *snapshots*. Each snapshot $s$ contains discrete signals $\mathbf{X}_{1:T}^s$ and the corresponding target $\mathbf{X}_{T+1}^s$ to predict.

### 6.2.3 Baselines and Target GNN Model

We select GLNN (Zhang et al., 2021), NOSMOG (Tian et al., 2023), and PPRGo (Bojchevski et al., 2020) as the baselines.

As for the target GNN model, we adopt the time-then-graph framework with the GRU-GCN model from Gao & Ribeiro (2022). For each $\mathbf{X}_{1:T}^s \in \mathbb{R}^{n \times T \times C}$, GRU-GCN uses a gated recurrent unit (GRU) to encode the signal at each node into the representation $\mathbf{H}^s \in \mathbb{R}^{n \times d} = \text{GRU}(\mathbf{X}_{1:T}^s)$, $d$ being the dimension of the representation. Then a graph neural network (GCN) is applied to $\mathbf{H}^s$ to obtain $\mathbf{Z}^s \in \mathbb{R}^{n \times d} = \text{GCN}(\mathbf{H}^s)$. Finally, an MLP decoder is used at each node to derive the prediction $\tilde{\mathbf{X}}_{T+1}^s = \text{MLP}(\mathbf{Z}^s) \in \mathbb{R}^{n \times C}$.

### 6.2.4 Evaluation

We split the snapshots into train, validation, and test sets. We report the test set's mean absolute percentage error (MAPE). We run each setting 10 times and report the mean and standard deviation of MAPE.

### 6.2.5 SDGNN Integration

We apply SDGNN to replace the GNN module. After training the GRU-GCN, we have GRU, GCN, and MLP decoder modules. For all the snapshots in the training set, we take $\mathbf{H}^s$ as the input to SDGNN and $\mathbf{Z}^s$ as the output to approximate. During training, we aggregate the loss and minimize over all snapshots:

$$\underset{\mathbf{\Theta}, \phi(\cdot; \mathbf{W})}{\text{minimize}} \quad \frac{1}{2} \sum_s \|\mathbf{\Theta}^\intercal \phi(\mathbf{H}^s; \mathbf{W}) - \mathbf{Z}^s\|_F^2 + \lambda_1 \|\mathbf{\Theta}\|_{1,1} + \lambda_2 \|\mathbf{W}\|_F^2,$$

We further train an MLP to map from the SDGNN output to the prediction. We also adopt various strategies to improve the robustness (please see the Appendix for details). During inference for node $z$, we refer to $\boldsymbol{\theta}_z$ to fetch the relevant node signals and apply the GRU. Then, we apply the SDGNN followed by the MLP to get the final prediction.

**Remark.** For scenarios with dynamic node features, we can rely on the GNN embeddings from past snapshots to train the SDGNN. We then apply the SDGNN to infer future snapshots. Due to the optimized and reduced number of receptive nodes, SDGNN can efficiently compute the prediction in the online prediction setting.

### 6.2.6 Main Results

Table 3 presents MAPE results under various settings. In this experiment, SDGNN's target model is GRU-GCN with a 2-layered GNN model. GRU denotes the degraded version of GRU-GCN that removes the GCN component. The GCN module integrates the information from neighbour nodes and reduces the error significantly. With the target GRU-GCN model, GLNN and NOSMOG slightly outperform the GRU, but the position encoding in NOSMOG does not help in this dynamic feature setting. SDGNN is closest to the target GRU-GCN model and performs best among the efficient models. SDGNN can select the most informative receptive nodes whose features have an impact on the labels. The heuristic-based neighbour selection method PPRGo fails under the current setting, achieving similar or even worse performance than GRU.

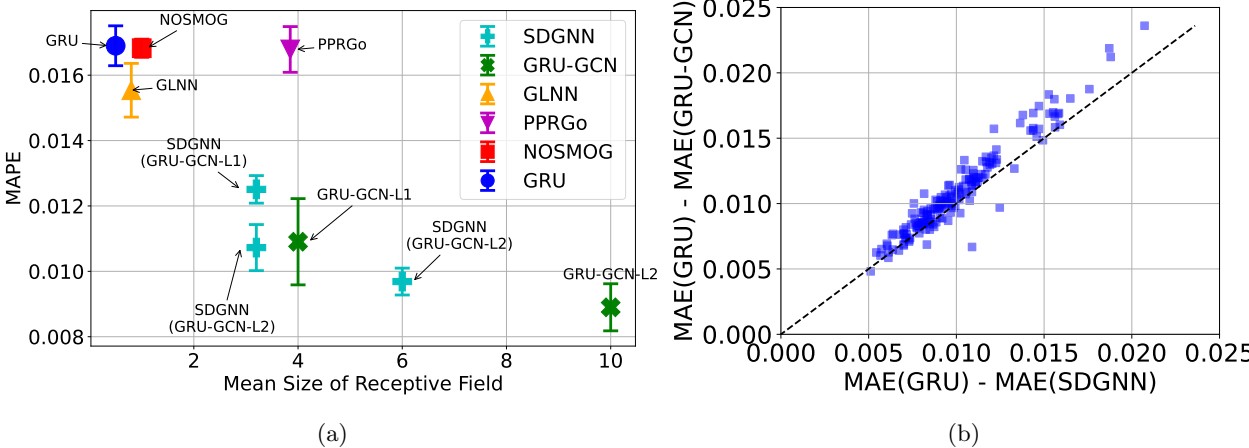

(a)                                                                (b)

Figure 3: (a) MAPE v.s. mean receptive field size on PeMS08 dataset. L1, L2 denote one and two GCN layers, respectively. GLNN and MOSMOG are learned with GRU-GCN-L2. Targets of SDGNN are indicated in brackets. Error bar shows standard deviation. (b) Scatter plot of MAE reduction at each node (compared to GRU) for GRU-GCN (y-axis) with SDGNN (x-axis). Dashed line is $y = x$. The improvements are highly correlated, with a slight bias in favour of the more computationally expensive GRU-GCN.

Figure 3a plots MAPE versus mean receptive field size for the PeMS08 dataset. We include the results of GRU-GCN models with 1 and 2 GCN layers. NOSMOG and SDGNN are trained with a GRU-GCN target of 2-layer GCN. SDGNN is trained in various settings for 1-layer and 2-layer GCN under different budgets of receptive field sizes. Although efficient, GLNN and NOSMOG have little performance gain over the GRU. SDGNN can effectively reduce the mean receptive field size with a slight accuracy sacrifice to the target GCN model. Comparing the SDGNN trained with different target GCN models, the SDGNN with 2-layer GCN is stronger than it trained with 1-layer GCN given a similar receptive field size. Training SDGNN using a more powerful target model with an aggressive reduction in receptive field size is more effective.

### 6.2.7 Embedding Approximation Efficacy

Figure 3b shows a scatter plot of the MAE reduction (relative to the GRU) at each node achieved by the GCN versus that achieved by SDGNN. The scatterplot is concentrated close to the $y = x$ line, with a slight bias in favour of the more computationally demanding GCN. The plot highlights how SDGNN successfully approximates the GCN embeddings and thus derives a similar forecasting accuracy improvement at each node.

## 7    Conclusion

In this work, we proposed a systematic approach to efficiently infer node representations from any GNN models while considering both the graph structures and the features from neighbour nodes. Extensive experiments on seven datasets on the node classification task and two datasets on the spatial-temporal forecasting task demonstrate that SDGNN can effectively approximate the GNN target models with reduced receptive field size. Our method provides a new tool for the efficient inference of GNN models under the online prediction setting, which can benefit from both the power of GNN models and the real-time reflection of dynamic node features.

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

# A Extended Experiment Settings on Node Classification Tasks

In this section, we provide additional details about our implementation to support the reproduction of our reported results. We ran all the tests on the computer equipped with Intel(R) Xeon(R) Gold 6140 CPU @ 2.30GHz CPU and NVIDIA Tesla V100 GPU.

## A.1 Dataset

Table 4 shows the detailed specifications of the datasets. The Arxiv and Products datasets are significantly larger than the other datasets.

Table 4: Summary of the Node Classification Datasets.

| Data | Nodes | Edges | Attributes | Classes |
|------|-------|-------|------------|---------|
| Cora | 2,708 | 5,278 | 1,433 | 7 |
| Pubmed | 19,717 | 44,324 | 500 | 3 |
| Citeseer | 3,327 | 4,552 | 3,703 | 6 |
| Computers | 13,752 | 245,861 | 767 | 10 |
| Photo | 7,650 | 119,081 | 745 | 8 |
| Arxiv | 169,343 | 1,166,243 | 128 | 40 |
| Products | 2,449,029 | 61,859,140 | 100 | 47 |

## A.2 Hyper-parameter Settings for Baselines

Table 5 shows the hyper-parameters for the base GNN models. Specifically, "lr" presents the learning rate; "hidden dimension" shows the hidden dimension of the intermediate layers; "layer" depicts the number of layers in the model; "weight decay" is the factor in the L2 regularizer for all learnable parameters. We adopt the same hyper-parameter of SAGE models in (Zhang et al., 2021). We adopt the hyper-parameters from those corresponding papers for geomGCN, Exphormer, DRGAT and RevGNN. Table 6 shows the hyper-parameters for the MLP student models for the GLNN, and Table 7 shows the hyper-parameters for the MLP student models for the NOSMOG. We adopt the same hyper-parameters from (Zhang et al., 2021) and (Tian et al., 2023) for the corresponding datasets. Table 8 shows the hyper-paramters for CoHOp.

Table 5: Summary of the hyper-parameters for base GNN models.

| Dataset/Model | lr | hidden dimension | layer | weight decay |
|---------------|-----|------------------|-------|--------------|
| Cora/SAGE | 0.01 | 64 | 3 | 0.0005 |
| Cora/geomGCN | 0.05 | 16 | 2 | 5.00E-06 |
| Citeseer/SAGE | 0.01 | 64 | 3 | 0.0005 |
| Citeseer/geomGCN | 0.05 | 16 | 2 | 5.00E-06 |
| Pubmed/SAGE | 0.01 | 64 | 3 | 0.0005 |
| Pubmed/geomGCN | 0.05 | 16 | 2 | 5.00E-06 |
| Computer/SAGE | 0.01 | 128 | 2 | 0.0005 |
| Computer/Exphormer | 0.001 | 80 | 4 | 0.001 |
| Photo/SAGE | 0.01 | 128 | 2 | 0.0005 |
| Photo/Exphormer | 0.001 | 64 | 4 | 0.001 |
| Arxiv/SAGE | 0.01 | 256 | 3 | 0.0005 |
| Arxiv/DRGAT | 0.002 | 256 | 3 | 0 |
| Products/SAGE | 0.001 | 256 | 3 | 0.0005 |
| Products/RevGNN | 0.003 | 160 | 7 | 0 |

### A.2.1 Hyper-parameter Setting for SDGNN

Table 9 shows the important hyper-parameters of SDGNN that yield the main results in Table 2. $K_1$ and $K_2$ are the hyper-parameters defined in the subsection, "Narrowing the Candidate Nodes". The $K_2$ fanout parameter specifies the neighbour sampling size. These three values jointly influence the sizes of the

Table 6: Summary of the hyper-parameters in the student MLP models for GLNN.

| Dataset/Model | lr | hidden dimension | layer | weight decay |
|---|---|---|---|---|
| Cora/SAGE | 0.005 | 128 | 2 | 0.0001 |
| Cora/geomGCN | 0.005 | 128 | 2 | 0.0001 |
| Citeseer/SAGE | 0.01 | 128 | 2 | 0.0001 |
| Citeseer/geomGCN | 0.01 | 128 | 2 | 0.0001 |
| Pubmed/SAGE | 0.005 | 128 | 2 | 0 |
| Pubmed/geomGCN | 0.005 | 128 | 2 | 0 |
| Computer/SAGE | 0.001 | 128 | 2 | 0.002 |
| Computer/Exphormer | 0.001 | 128 | 2 | 0.002 |
| Photo/SAGE | 0.005 | 128 | 2 | 0.002 |
| Photo/Exphormer | 0.005 | 128 | 2 | 0.002 |
| Arxiv/SAGE | 0.01 | 256 | 3 | 0 |
| Arxiv/DRGAT | 0.01 | 256 | 3 | 0 |
| Products/SAGE | 0.01 | 256 | 4 | 0.0005 |
| Products/RevGNN | 0.01 | 256 | 4 | 0.0005 |

Table 7: Summary of the hyper-parameters in the student MLP models for NOSMOG.

| Dataset/Model | lr | hidden dimension | layer | weight decay |
|---|---|---|---|---|
| Cora/SAGE | 0.01 | 128 | 2 | 0.0001 |
| Cora/geomGCN | 0.01 | 128 | 2 | 0.0001 |
| Citeseer/SAGE | 0.01 | 128 | 2 | 1.00E-05 |
| Citeseer/geomGCN | 0.01 | 128 | 2 | 1.00E-05 |
| Pubmed/SAGE | 0.003 | 128 | 2 | 0 |
| Pubmed/geomGCN | 0.003 | 128 | 2 | 0 |
| Computer/SAGE | 0.001 | 128 | 2 | 0.002 |
| Computer/Exphormer | 0.001 | 128 | 2 | 0.002 |
| Photo/SAGE | 0.001 | 128 | 2 | 0.001 |
| Photo/Exphormer | 0.001 | 128 | 2 | 0.001 |
| Arxiv/SAGE | 0.01 | 256 | 3 | 0 |
| Arxiv/DRGAT | 0.01 | 256 | 3 | 0 |
| Products/SAGE | 0.003 | 256 | 3 | 0 |
| Products/RevGNN | 0.003 | 256 | 3 | 0 |

candidate sets, and hence the training time of SDGNN and the approximation quality. We conduct a grid search over $K_1 = [1, 2, 3]$ and $K_2 = [1, 2]$, starting from the combination that results in smaller candidate sets to larger ones. We select the combination such that the associated training time of SDGNN is feasible, and the validation loss in (3) does not reduce by more than 0.01 compared with the next smaller setting.

Table 8: Summary of the hyper-parameters for CoHOp. If a hidden dimension is not stated, a linear model is used.

| Dataset | hidden dimension | # epochs | patience | tau | gamma | lr | weight decay | alpha |
|---|---|---|---|---|---|---|---|---|
| Cora | - | 200 | 50 | 0.5 | 0.025 | 0.1 | 0.0005 | 0.4 |
| Citeseer | - | 200 | 50 | 0.6 | 0.005 | 0.1 | 0.0005 | 0.5 |
| Pubmed | - | 200 | 50 | 0.2 | 0.01 | 0.1 | 0.0001 | 0.6 |
| Computer | - | 200 | 50 | 0.2 | 0.01 | 0.01 | 0.0 | 0.05 |
| Photo | - | 200 | 50 | 0.4 | 0.05 | 0.01 | 0.0005 | 0.1 |
| Arxiv | 512 | 200 | 100 | 0.5 | 0.0 | 0.1 | 0.001 | 0.6 |
| Products | 1024 | 1 | 1 | 0.5 | 0.5 | 0.001 | 0.0 | 0.1 |

Table 9: Summary of the Hyper-parameters for Training SGDNN.

| Data/Target model | $K_1$ | $K_2$ | $K_2$ fanout | lr | $l_2$ |
|---|---|---|---|---|---|
| Cora/SAGE | 2 | 2 | 10, 10 | 0.01 | 1e-05 |
| Cora/geomGCN | 2 | 2 | 10, 10 | 0.001 | 1e-07 |
| Citeseer/SAGE | 2 | 2 | 10, 10 | 0.001 | 1e-05 |
| Citeseer/geomGCN | 2 | 2 | 10, 10 | 0.001 | 1e-06 |
| Pubmed/SAGE | 2 | 1 | 10 | 0.001 | 1e-05 |
| Pubmed/geomGCN | 2 | 1 | 10 | 0.001 | 1e-06 |
| Computer/SAGE | 1 | 2 | 10, 10 | 0.01 | 1e-05 |
| Computer/exphormer | 1 | 2 | 10, 10 | 0.005 | 1e-05 |
| Photo/SAGE | 1 | 2 | 10, 10 | 0.01 | 1e-05 |
| Photo/exphormer | 1 | 2 | 10, 10 | 0.001 | 1e-05 |
| Arxiv/SAGE | 1 | 2 | 15, 15 | 0.001 | 1e-05 |
| Arxiv/DRGAT | 1 | 2 | 15, 15 | 0.0001 | 1e-07 |
| Products/SAGE | 1 | 2 | 10, 10 | 0.001 | 1e-05 |
| Products/RevGNN | 1 | 2 | 10, 10 | 0.0005 | 1e-07 |

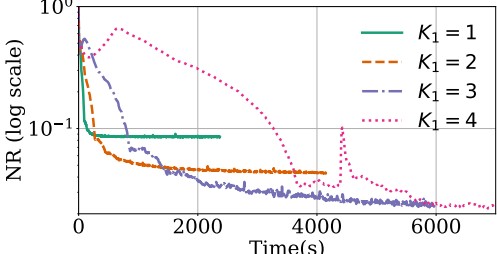

Figure 4: Convergence curve under various candidate sets for Pubmed dataset with SAGE model.

# B  Additional Experimental Results for Node Classification Tasks

## B.1  The Impact of Candidate Selection

We train the SDGNN with different settings for selecting the candidate sets and show the *normalized regret* (NR) v.s. the wall clock time. The normalized regret is defined as

$$\mathrm{NR} = \frac{1}{2|\mathcal{V}|} \sum_{z \in \mathcal{V}} \frac{\|\boldsymbol{\theta}_z^\mathsf{T} \phi(\mathbf{X}; \mathbf{W}) - \boldsymbol{\Omega}_{z*}\|_2^2}{\|\boldsymbol{\Omega}_{z*}\|_2^2},$$

which characterize the average relative error with respect to the target GNN embeddings. Fig. 4 shows the training curve for the Pubmed dataset with the SAGE model under four different settings of candidate sets, i.e., including 1-hop, 2-hop, 3-hop or 4-hop neighbours, respectively. As we can see, the smaller candidate sets, e.g. $k_1 = 1$ or $k_1 = 2$, leads to a faster convergence but converges to a sub-optimal point, while a large candidate set converges much slower with a better result. With $k_1 = 4$, the normalized regret even bounces back and forth. Besides, if we adopt all the nodes as candidates, the training will take an extremely long time (not shown in the figure). We conclude that with a proper setting of the candidate set, the learning of SDGNN can be greatly boosted while losing little performance in terms of NR.

## B.2  The Receptive Nodes from SDGNN

Table 10 presents the average receptive field sizes for SDGNN and the original graph. It also reports their ratios. To make an efficient estimation, we compute the mean statistics from 1000 randomly sampled nodes. For the original graph, we only count up to 2-hop neighbours for Products, due to the very rapid expansion of the neighbourhood. We count up to 3-hop neighbours for the other scenarios.

SDGNN can significantly reduce the receptive field size against the original graph. Interestingly, the ratio between the receptive field size of SDGNN and that from the original graph generally decreases as the hop size increases. This indicates that, proportionally, SDGNN favours receptive nodes that are closer to the center node. However, in each case, the majority of the nodes that SDGNN selects are not immediate neighbours.

Table 10: The average number of receptive nodes for each node ("Overall"); the average number at a distance of "1-hop", "2-hops" and "3-hops" for (i) SDGNN, (ii) original graph (iii) the ratio between SDGNN and original graph.

| Data/Model | Overall | | | 1-hop | | | 2-hop | | | 3-hop | | |
|---|---|---|---|---|---|---|---|---|---|---|---|---|
| | SDGNN | 3-hop Neigh. | Ratio | SDGNN | Original | Ratio | SDGNN | Original | Ratio | SDGNN | Original | Ratio |
| cora/SAGE | 36.1 | 128.1 | .28 | 2.9 | 4.9 | .59 | 10.1 | 31.7 | .32 | 12.4 | 92.9 | .13 |
| cora/geomGCN | 38.4 | 128.1 | .30 | 2.2 | 4.9 | .45 | 10.2 | 31.7 | .32 | 13.6 | 92.9 | .15 |
| citeseer/SAGE | 17.8 | 43.5 | .41 | 2.7 | 3.9 | .69 | 4.7 | 12.1 | .39 | 6 | 29.8 | .20 |
| citeseer/geomGCN | 14.2 | 43.5 | .33 | 1.8 | 3.9 | .46 | 3.5 | 12.1 | .29 | 5 | 29.8 | .17 |
| pubmed/SAGE | 25.5 | 394.6 | .06 | 2.6 | 5.7 | .46 | 13.1 | 54.8 | .24 | 9.7 | 336.6 | .03 |
| pubmed/geomGCN | 16.3 | 394.6 | .04 | 1.1 | 5.7 | .19 | 8 | 54.8 | .15 | 7.2 | 336.6 | .02 |
| a-computer/SAGE | 24.9 | 7493 | .003 | 2.6 | 37.3 | .07 | 15 | 1906.8 | .008 | 7.4 | 5727.6 | .001 |
| a-computer/exphormer | 46.9 | 7493 | .006 | 5.3 | 37.3 | .14 | 28.4 | 1906.8 | .01 | 13.4 | 5727.6 | .002 |
| a-photo/SAGE | 28.1 | 2519.4 | .01 | 4.1 | 29 | .14 | 17.4 | 759.3 | .02 | 6.1 | 1689.8 | .003 |
| a-photo/exphormer | 45.8 | 2519.4 | .018 | 5.9 | 29 | .2 | 28 | 759.3 | .04 | 11.4 | 1689.8 | .007 |
| ogbn-arxiv/SAGE | 30.3 | 18470.6 | .0016 | 2 | 14 | .14 | 12.9 | 3713 | .003 | 16 | 15443.1 | .001 |
| ogbn-arxiv/DRGAT | 36.1 | 18470.6 | .0019 | 2.3 | 14 | .16 | 14.2 | 3713 | .004 | 20 | 15443.1 | .001 |
| ogbn-products/SAGE | 24 | 3851.9 | .006 | 2.1 | 51.9 | .04 | 11.4 | 3919.2 | .003 | 10.7 | | |
| ogbn-products/RevGNN-112 | 23.6 | 3851.9 | .006 | 1.8 | 51.9 | .03 | 10.3 | 3919.2 | .003 | 11.6 | | |

Figure 6 and Figure 5 show the empirical CDFs (calculated from 1000 randomly sampled center nodes) of the number of receptive nodes and the theta values (node weights after $l_1$ row normalization) for SDGNN, SGC and PPRGo. For SGC and PPRGO, we include all the weights without truncating. For SGC, we include the 3-hop neighbours for all datasets except for Products, which includes only the 2-hop neighbours. SGDNN consistently identifies a smaller number of receptive nodes and puts larger weights on a few focused nodes.

This is the main reason that SDGNN can significantly reduce the receptive field size. Although PPRGo often selects a similar number of receptive nodes as SDGNN, it distributes the weights more evenly among them.

## B.3 Inference Time

To fairly compare the inference times, we always constrain the computation on a single CPU and compute the prediction of 1 node at a time. This simulates the online prediction setting where the request comes randomly. We show the inference times over 10000 randomly sampled testing nodes for all the datasets in Table 11. Specifically, the MLP column presents the mean inference time for a 3-layered MLP model, and SAGE-N20 presents the mean inference time of the SAGE models with a neighbour sampling size of 20. The SDGNN-SAGE and SDGNN-SOTA columns report the inference times of SDGNN trained with the SAGE model and the corresponding SOTA model in Table 2. We can see that SDGNN consistently has a much smaller inference time compared to the SAGE counterpart, and it is on the same scale as the MLP models (inference time is 1–4× larger).

Table 11: The average inference time per node over 10000 samples. (ms)

| Dataset | MLP | SAGE-N20 | SDGNN-SAGE | SDGNN-SOTA |
|---|---|---|---|---|
| cora | 0.2 | 11.5 | 0.5 | 0.8 |
| citeseer | 0.3 | 11 | 0.6 | 0.9 |
| pubmed | 0.2 | 10.8 | 0.3 | 0.5 |
| a-computer | 0.2 | 44.4 | 0.6 | 0.4 |
| a-photo | 0.2 | 7.4 | 0.6 | 0.4 |
| ogbn-arxiv | 0.2 | 15.6 | 0.2 | 0.4 |
| ogbn-products | 0.4 | 13.6 | 0.7 | 0.7 |

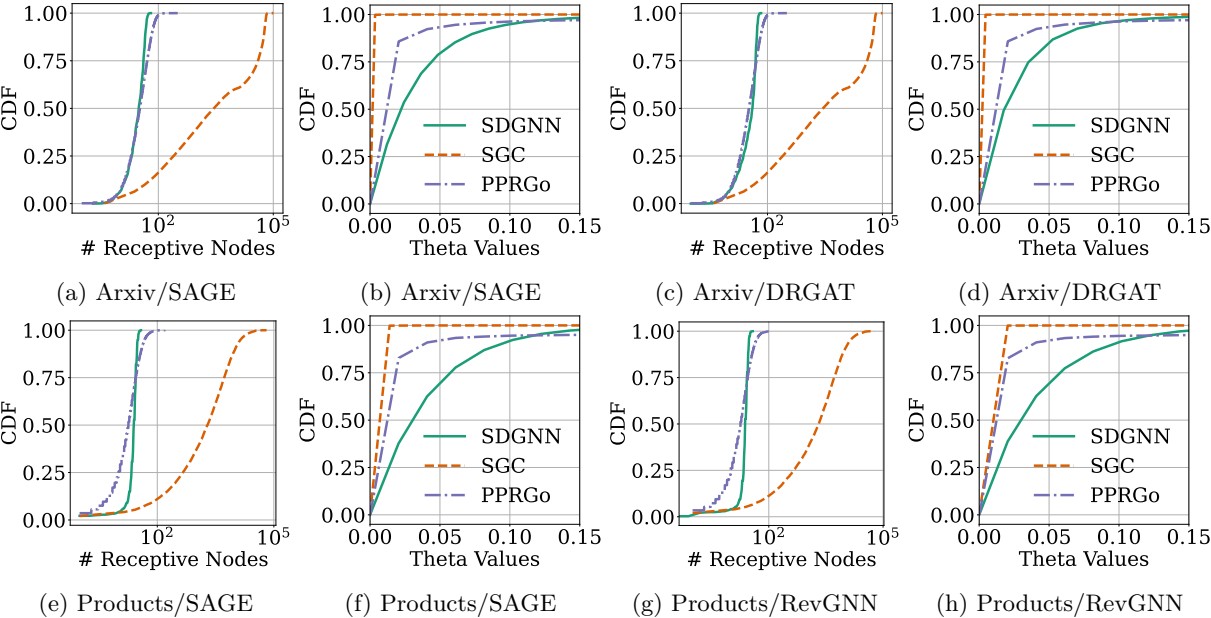

Figure 5: The empirical CDFs of the number of receptive nodes and the empirical CDFs of the row normalized $\Theta$ for SDGNN, SGC and PPRGo for ArXiv (SAGE, DRGAT) and Ogbn-Products (SAGE, RevGNN-112).

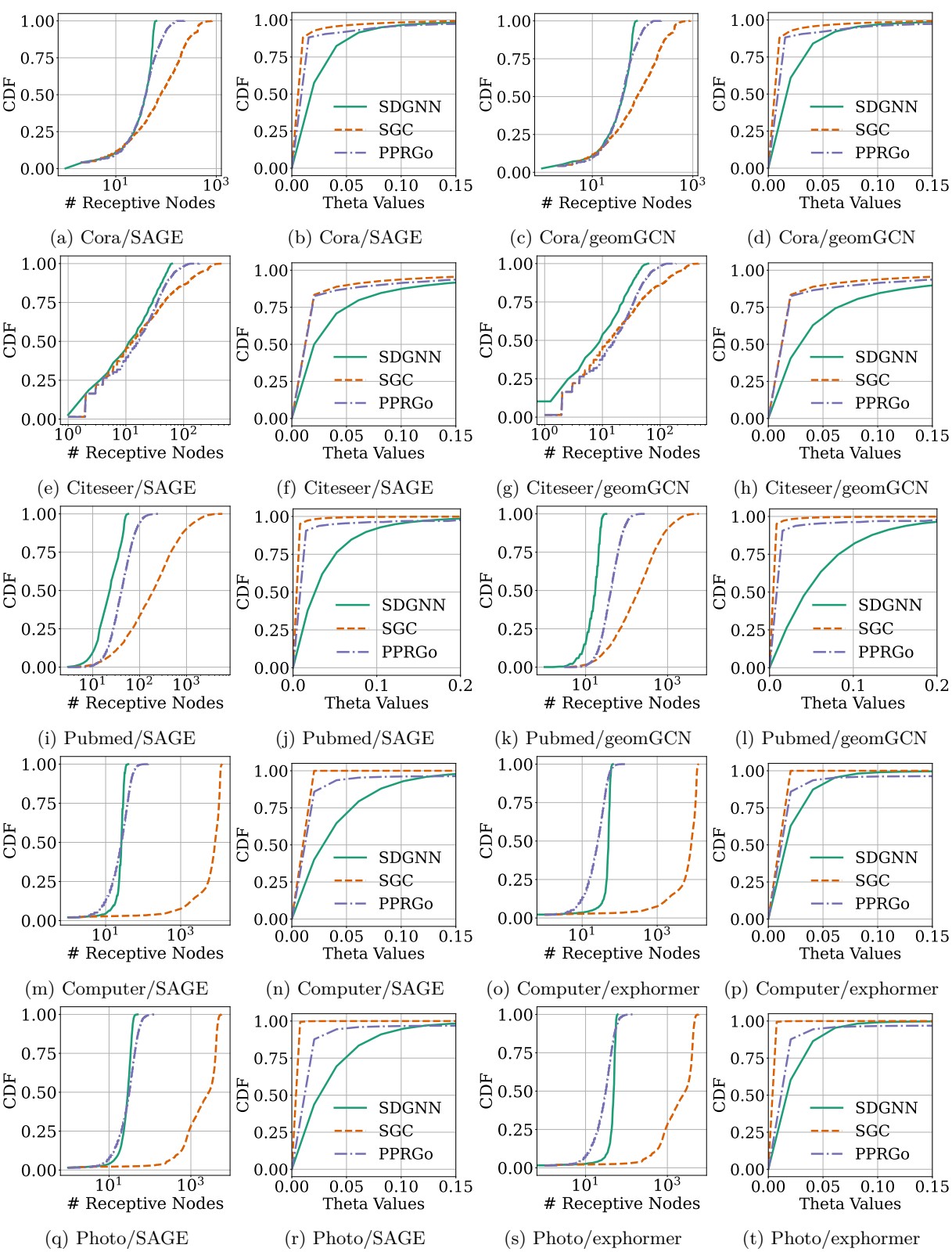

Figure 6: The empirical CDFs of the number of receptive nodes and the empirical CDFs of the row normalized $\Theta$ for SDGNN, SGC, PPRGo for Cora, Citeseer and Pubmed (SAGE, geomGCN), and Computer and Photo (SAGE, exphormer).

## C  Implementation Details for GRU-GCN and SDGNN and Analysis in Spatio-temporal Tasks

The implementation of the target model for the spatio-temporal task follows the methodology outlined by (Gao & Ribeiro, 2022). We strictly adhered to the implementation details of the GRU-GCN model as described in that work. The overall architecture of GRU-GCN is illustrated at the top of Figure 7. It consists of three main components: an RNN, a GCN, and a final MLP. The hidden dimensions for the RNN, GCN, and MLP components are identical with a softplus as the activation function. The target GRU-GCN model was trained for 1000 epochs, with early stopping applied using a patience of 30 epochs. The hyperparameters used for the target model are summarized in Table 12.

Figure 7 (bottom) illustrates what we employed for integrating SDGNN in the spatio-temporal setting. We directly adopted the trained RNN from the GRU-GCN model, followed by the SDGNN trained using the embeddings from the GCN. This was then followed by a fine-tuned MLP, originally from the GRU-GCN model. Ultimately, SDGNN consists of two components: an MLP and sparse weights. After obtaining the target model, we trained SDGNN for 4000 epochs, employing an iterative training and optimization process for the MLP and sparse weights. The sparse weights were optimized every 40 epochs. Following the training of SDGNN, we fine-tuned the MLP for the low-level task over 30 epochs. The best-performing SDGNN and low-level task MLP were selected based on their MAPE performance on the validation dataset. In a spatio-temporal context, the task is essentially a regression problem. Unlike classification tasks, and due to our method's focus on approximating the final graph embedding, we observed that the last projection layer or the final MLP is sensitive to differences between the approximated embeddings and the target GNN embeddings. To enhance the robustness of the final MLP layer in the GRU-GCN model, we introduced noise between the GCN and MLP during training. This noise was carefully calibrated as a hyperparameter to avoid degrading the overall performance of the GRU-GCN while still providing the necessary robustness to the final low-level task MLP. Specifically, we added Gaussian random noise $\mathcal{N}(0, 0.05)$ for the PeMS08 dataset and $\mathcal{N}(0, 0.01)$ for the PeMS04 dataset. The detailed hyperparameters for SDGNN are presented in Table 13.

Table 12: Sets of hyperparameters for GRU-GCN in spatio-temporal setting experiment.

| Dataset/Model | lr | hidden dim. | GCN layer | MLP layer | weight decay | patience |
|---|---|---|---|---|---|---|
| **PeMS04** | 1e-3 | 16 | 2 | 2 | 1e-5 | 30 |
| **PeMS08** | 1e-3 | 16 | 2 | 2 | 1e-5 | 30 |

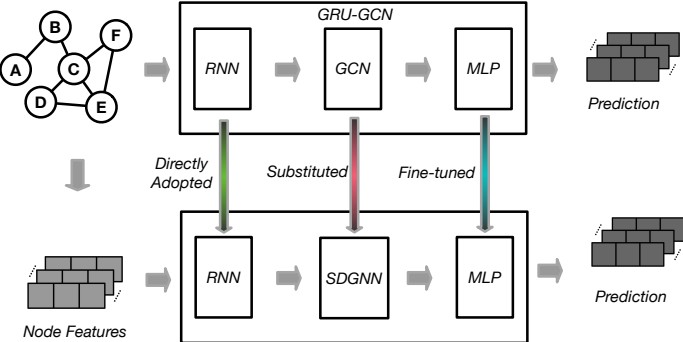

Figure 7: The simplified architecture diagram of the GRU-GCN model is depicted at the top. Building on the GRU-GCN architecture, we substitute the GCN module with SDGNN to enhance inference efficiency. For a comprehensive overview of the SDGNN pipeline, please refer to Figure 1.

## C.1 Optimization Schedule for the Sparse Weights

To ensure that the weights remain sparse, we implemented a specific optimization schedule tailored to the characteristics of sparse weights. At the beginning of the training process, we set a larger maximum receptive field size to allow for broader connections. As training progresses, we gradually reduce the maximum receptive field size by 1 every 100 epochs. In our case, the largest receptive field size at the beginning of the training was set to 46. Throughout 4000 training epochs, we gradually reduced the receptive field size. Specifically, every 100 epochs, the maximum receptive field size was decreased by 1, resulting in a final receptive field size of 6.

Table 13: Sets of hyperparameters for SDGNN in spatio-temporal setting experiment. Underlined values are those selected by grid search.

| Parameters | hidden dim. | num. layers | noise level | lr |
|---|---|---|---|---|
| **PeMS04** | {8,16,32} | {1, 2, 4} | {0.01, 0.05, 0.1, 0.15} | {0.001, 0.0025, 0.005, 0.01} |
| **PeMS08** | {8,16,32} | {1, 2, 4} | {0.01, 0.05, 0.1, 0.15} | {0.001, 0.0025, 0.005, 0.01} |

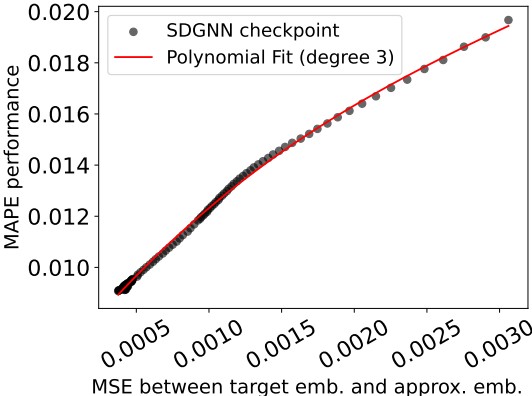

Figure 8: Prediction performance in MAPE relative to the MSE between SDGNN approximated embeddings and target model embeddings in the spatio-temporal setting

## C.2 Analysis of Final Performance Relative to the Distance Between SDGNN Approximated Embeddings and Target Model Embeddings

Since SDGNN aims to approximate the target model embeddings, we hypothesize that the discrepancy between the SDGNN embeddings and the target model embeddings will significantly impact the overall model performance. Figure 8 illustrates the relationship between the mean squared error (MSE) of the embeddings and the prediction performance. The x-axis represents the MSE between the target embeddings and the SDGNN-approximated embeddings, while the y-axis indicates the prediction performance measured by the MAPE metric. Each scatter point represents a checkpoint of SDGNN. We utilize checkpoints of SDGNN during training at every 100 epochs and evaluate the performance on the test sets to generate the plot. The figure reveals that higher approximation errors correspond to greater discrepancies in prediction performance. This observation suggests that as the MSE between the SDGNN embeddings and the target embeddings decreases, the model's performance is likely to improve.

## C.3 Analysis on Performance Degradation with Increasing Model Staleness

In this experiment, we analyze the prediction error between the ground truth and predicted values by calculating the absolute error for each data point. The testing set is divided into 10 equal-sized chunks, preserving the chronological order of the data. Figure 9 presents the mean absolute error with standard deviation for every chuck of the test set for both PeMS04 and PeMS08 datasets. We can see that the

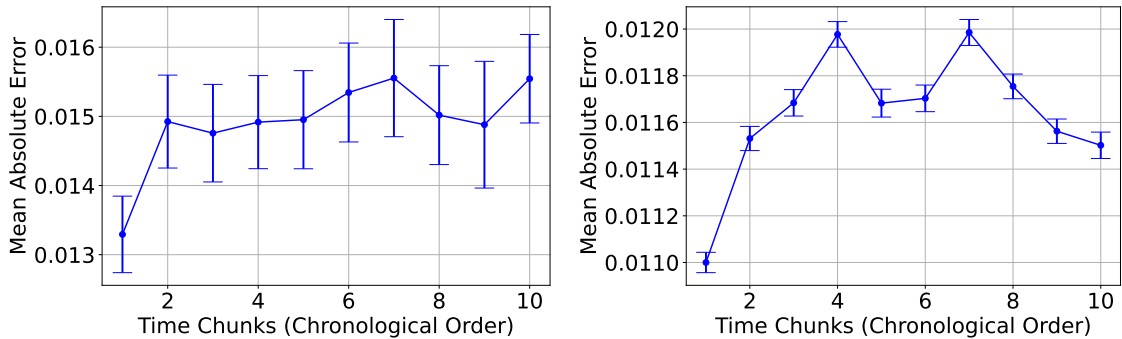

Figure 9: The mean absolute error per chronological chunk showing potential shifts in prediction accuracy over time for PeMS04 (Left) and PeMS08 (Right) datasets. Error bars represent the standard deviation within each chunk.

performance indeed degrades with later time chunks, but the degradation quickly saturated at a slightly worse point than in the first chunk.

## D  Additional Analysis on Expressive Power of SDGNN

The analysis for a general GNN target is challenging. We demonstrate the expressive of SDGNN by showing how SDGNN can approximate any Filter-Most-Expressive (FME) GNNs defined in (Wang & Zhang, 2022), which covers a wide range of spectral GNNs. Let $\mathbf{A}$ denote the adjacency matrix and $\mathbf{D}$ denote the diagonal matrix where $\mathrm{D}_{ii}$ denotes the degree of node $i$. The normalized Laplacian matrix $\mathbf{L} := \mathbf{I} - \mathbf{D}^{-1/2}\mathbf{A}\mathbf{D}^{-1/2}$ where $\mathbf{I}$ is the identity matrix. Let $\mathbf{L} = \mathbf{U}\mathbf{\Lambda}\mathbf{U}^{\intercal}$ denote the eigendecomposition. Let the function $\beta(\mathbf{\Lambda})$ applies $\beta$ element-wisely to the diagonal matrix $\mathbf{\Lambda}$. We further denotes $\tilde{\beta}(\mathbf{\Lambda}) = \mathbf{U}\beta(\mathbf{\Lambda})\mathbf{U}^{\intercal}$. Then, the FME GNN is defined as

$$g_{\mathrm{FME}}(\mathbf{X}) := \alpha(\mathbf{U}\beta(\mathbf{\Lambda})\mathbf{U}^{\intercal}\varphi(\mathbf{X})) = \alpha(\tilde{\beta}(\mathbf{\Lambda})\varphi(\mathbf{X})), \tag{10}$$

where $\mathbf{X} \in \mathbb{R}^{|\mathcal{V}| \times D}$ is the node features, $\alpha(\cdot)$ and $\varphi(\cdot)$ are functions like multi-layer perceptrons that operate row-wisely on the input matrix and $\beta(\cdot)$ is arbitrary real-valued filter function. Consider the challenging case that $\mathbf{X}$ is a random variable drawn from a certain distribution. We want to show that under mild conditions, there exists an SDGNN equipped with some function $f(\cdot)$, such that the $f(\mathbf{\Theta}^{\intercal}\phi(\mathbf{X}; \mathbf{W}))$ is close to $g_{\mathrm{FME}}(\mathbf{X})$, for $\forall \mathbf{X}$ drawn in such a distribution.

We set $f(\cdot)$ to be $\alpha(\cdot; \mathbf{W})$ and $\phi(\cdot)$ to be $\varphi(\cdot)$, respectively. Assuming $\alpha(\cdot)$ is Lipschitz continuous, we only need to show that there exists some $\mathbf{\Theta}$, such that $\mathbf{\Theta}^{\intercal}\varphi(\mathbf{X})$ is close to $\tilde{\beta}(\mathbf{\Lambda})\varphi(\mathbf{X})$. A trivial solution without considering sparsity is to take $\mathbf{\Theta}^{\intercal} = \tilde{\beta}(\mathbf{\Lambda})$, and $f(\mathbf{\Theta}^{\intercal}\phi(\mathbf{X}; \mathbf{W}))$ can perfectly express $g_{\mathrm{FME}}(\mathbf{X})$. With sparsity constraint, we can treat it as solving a Lasso problem to decide each column of $\mathbf{\Theta}$. Then, there is a trade-off between the quality and sparsity. Specifically, for the $n$th column of $\mathbf{\Theta}$, if we set the number of non-zero entries to be $k$, the quality depends on the smallest distance between the $n$th row of $\tilde{\beta}(\mathbf{\Lambda})\varphi(\mathbf{X})$ and all the subspaces spanned by $k$ rows of $\varphi(\mathbf{X})$. Such quality monotonically increases if we allow larger $k$. If most rows of $\tilde{\beta}(\mathbf{\Lambda})\varphi(\mathbf{X})$ are close to some subspaces spanned with small $k$, there exists a parameter setting of SDGNN with sparse $\mathbf{\Theta}$ that can approximate $g_{\mathrm{FME}}(\mathbf{X})$.

## E  Limitations of SDGNN

Although SDGNN is very effective in approximating target GNN embeddings, there are a few limitations. We present the naive cues from an engineering perspective and would like to explore principled solutions in future works.

### E.1 Incompatibility with Inductive Setting

SDGNN always requires the GNN embeddings from all nodes as the guiding signal to generate the optimal $\boldsymbol{\theta}$ and $\phi$. Therefore, SDGNN cannot be applied directly in the inductive setting when new nodes may appear during inference time. From the application perspective, a simple strategy to handle the novel nodes in real-time could be collecting the $\boldsymbol{\theta}$ for all/sampled 1-hop neighbour nodes and summing them up as the proxy for the new nodes. Moreover, for the situation where we are allowed to do some preprocessing for the novel nodes, we could use the inductive GNN to infer the GNN embedding, compute $\boldsymbol{\theta}$ with Phase $\boldsymbol{\theta}$ for that node and cache it. Then, the actual online inference stage will be the same as the other nodes that are seen during the training of SDGNN.

### E.2 Unadaptability to Distribution Shift

For the online prediction setting, we train SDGNN based on past snapshots and apply it in future snapshots. We did not consider the potential distribution shift between training and testing. If the time gap between the training and the testing is large, such a shift might degrade the performance.

A typical and practical solution is periodically retraining new models based on recently collected data. This can solve the distribution shift issue for both the target GNN model and SDGNN. If we trade the performance for less computation overhead, we can periodically and partially update SDGNN for better training efficiency. For example, we can keep the weight matrix $\mathbf{W}$ unchanged and only periodically perform a Phase $\boldsymbol{\Theta}$ update according to the latest data.

A fundamental approach could incorporate online learning so that the trainable weights from the target GNN and SDGNN can be adapted as more data become available and the graph topology changes. However, developing such a method is involved due to the challenges of enforcing a sparse solution. We would leave this as a future work.

### E.3 Inability to Represent Non-linear Interactions between the Features at Different Nodes

In standard GNN models, the recursive procedure of aggregating neighbour features followed by the feature transformation may potentially model the non-linear feature interaction between nodes. However, SDGNN is formulated as a linear combination of transformed node features. Although efficient, it seems not to have the potential to model arbitrary feature interaction between nodes. One immediate fix is that for any target node, we can augment the other candidate nodes' features with the target node feature before feeding into $\phi$, which appears to integrate all the feature interaction between the neighbour nodes and the target nodes. However, a general principled approach is worth exploring. Besides, we should have a theoretical analysis of the expressiveness of SDGNN and explore under which conditions SDGNN is not as expressive as the general GNN models.

### E.4 Training Overhead and Challenge on Hyper-parameter Selection

We managed to finish the training of SDGNN in about 20 hours for the graph with millions of nodes like OBGN-products, but training SDGNN on larger graphs is challenging. Regarding the hyper-parameter selection, such a training overhead for each run is a nightmare. The main obstacle is the solver for the Lasso problem. Implementing the LARS solver that supports the GPU speedup could be an immediate workaround. As for a principled solution, since we iteratively optimize $\boldsymbol{\theta}$ and $\phi$, we don't necessarily require an optimized result for Phase $\boldsymbol{\theta}$ at each iteration. Instead, we may propose a similar Lasso solver that can rapidly trade-off between the computation complexity and the accuracy of the solution. During the training of SDGNN, we could engage a schedule to gradually switch the Lasso solver from efficiently generating intermediate results to accurately achieving optimal results.

