# OpenReview forum: "Sparse Decomposition of Graph Neural Networks"
_TMLR — Accepted by TMLR_

### Review · Reviewer_M5hg · 2024-11-19

**Summary Of Contributions:**

The authors introduce SDGNN, an MLP-based method for approximating GNN embeddings that is linear w.r.t. average node degree and the number of layers in the GNN. Subsuming prior work with heuristic decoupling and knowledge distillation, SDGNN is novel in its approach by alternating between training an MLP to approximate the teacher GNN embeddings and optimizing sparse weights. This enables fast inference where each node embedding is computed as a linear combination of the other embeddings using those sparse weights. Through thorough experiments on graph and temporal graph node classification benchmarks, SDGNN outperforms existing methods with superior inference efficiency.

**Audience:**

Yes

**Claims And Evidence:**

Yes

**Requested Changes:**

* Can the authors elaborate on a more intuitive explanation for what SDGNN boils down to on an intuitive way? It seems SDGNN is essentially a way of conducting weighted message passing with teacher GNN supervision and sparse edge weights, but a clear intuition behind the method would be helpful given the difficult interpretability of sparse weights.
* Can the authors elaborate on the importance of each component of SDGNN on downstream performance/inference time? For instance, in Appendix D.4, the authors note that different strategies in optimizing the sparse weights can result in tradeoffs between computation complexity and accuracy. Further, how much do hyperparameters such as max LARS iterations and $K_1$ and $K_2$ affect performance?
* Can the authors elaborate on what the significance of dynamic node features with the shortcomings of existing GNN/MLP models in fast dynamic node inference? This advantage is mentioned in the beginning sections, but the explicit connection between dynamic node features and the temporal graph benchmark as well as explaining why SDGNN is superior in this regard could use more detail.

**Strengths And Weaknesses:**

**Strengths**

* **Temporal graph benchmarks**:  MLP-based graph GNN approximators are usually not benchmarked on temporal graph datasets, so it is a pleasant comparison to see how models perform in a different setting than the usual graph
* **General formulation**: SDGNN has the benefit of being more general than what the authors describe as "heuristic decoupling" approaches, and it's clear how its weights can be adjusted to subsume PPR/normalized adjacency matrix methods. This introduces new optimization challenges, which adds to the novelty of the paper.
* **Comprehensive experiments**: Experiments and results are very well-presented, and significance testing is used in the main results. All hyperparameters and optimization schedules are transparently described in detail in the Appendix.

**Weaknesses**

* **Missing baselines/benchmarks**: The authors are missing comparison with VQGraph [1], a new SOTA GNN-MLP distillation technique. I think comparison with this method would be beneficial. Also, Table 2 only shows results for homophilic graph datasets. Performance on heterophilic datasets with heterophily-minded GNN teachers would be helpful for evaluating SDGNN.
* **Lack of asymptotic analysis**: Especially for training SDGNN, the paper would benefit from an explicit analysis of the time complexity of SDGNN taking into account both the candidate node generation algorithm and the LARS solver.
* **Lack of ablation/sensitivity studies**: There are several hyperparameters that are important for SDGNN's performance, such as $K_1$, $K_2$, the number of LARS iterations, $\lambda_1$, and $\lambda_2$. It would be helpful to know how SDGNN performs differently w.r.t. different settings of these parameters.
* **Lack of training time cost** SDGNN requires both training a neural network with gradient descent and solving an optimization problem with a LARS solver. It seems SDGNN's training procedure can be quite expensive, requiring training for 4000 epochs alternating between training the MLP and the sparse weights. A discussion on the time and space costs of training SDGNN complete with a study examining how each component affects training/inference time would be helpful. As it stands, it seems SDGNN should be no cheaper to train than models like SAGE with few layers and small neighborhood size < $K_1K_2$.

[1] VQGraph: Rethinking Graph Representation Space for Bridging GNNs and MLPs

---

> ### Author Response · Authors · 2025-01-04
> **Reply for Comments Part 1: the Requested Changes**
>
> We sincerely appreciate your constructive and inspiring feedback.
>
> __Intuition of SDGNN.__ Rather than inventing an efficient model with great performance, we aim to find a fundamental and model-agnostic solution to approximate the prediction of any target GNN model with efficient computation at the inference stage. Some recent GNN-MLP works like NOSMOG and VQGraph not only learn from the original target GNN model but also implicitly impose some additional modules to boost the performance. Therefore, they cannot be guaranteed to achieve performance similar to that of the stronger target GNNs, which is empirically verified in Table 2. Since we focus on approximating any target GNNs, we introduce additional flexibility in the sparse weights for a potentially powerful and universal structure to mimic the performance of ANY GNNs with efficient inference. We will add more elaborations to clarify the motivation in a revised version.
>
> __Importance of components of SDGNN.__ Both the structure and the optimization algorithm of SDGNN are critical to the performance. The SDGNN will not work without any of them. In the structure of SDGNN, the learnable feature transformation function and the sparse aggregation weights grant the model sufficient flexibility to approximate a wide range of target GNN models. The optimization algorithm actually finds the optimal learnable parameters. As for the inference time, the dominating factor is the number of non-zero entries in the sparse weight vectors, which is mainly controlled by the max LARS iterations. The performance will generally increase if we enlarge the max LARS iterations, but the improvement will quickly saturate once the max LARS iterations reach some threshold. $K_1$ and $K_2$ are hyper-parameters that shrink the search space of the LARS algorithm to reduce the training time. The training time will be shorter as the search space decreases, but the performance might drop if the candidate lacks sufficient critical neighbours. We will add more elaborations on the impact of the components of SDGNN in a revised version.
>
> __Significance of dynamic node features.__ Besides the dynamic node features, SDGNN is also designed to handle the dependency on neighbour features. These features are valuable but challenging to achieve. First, some applications could benefit from such features. For example, the recent purchasing behaviour of one's closest friend might probably influence her online shopping preference; the traffic flow of an intersection in the near future should be correlated with the status of its neighbouring intersections, etc. Therefore, we want to develop a technique to efficiently summarize or approximate the dynamic and potentially critical neighbour features to fully convey the power of GNNs. Second, achieving these features simultaneously is challenging. Without dynamic node features, we always have an easy workaround method to cache all the predictions instead of cumbersome computation upon request. Without the critical dependency on neighbour features, the GNN-MLP approaches should work well. However, none of the existing methods can deal with the case where prediction could benefit from potentially dynamic node features, especially for those from a few critical neighbours. We will add more elaboration with examples to better convey the motivation of this work.

---

> ### Author Response · Authors · 2025-01-04
> **Reply for Comments Part 2: the Weaknesses**
>
> __Compare with VQGraph.__ Thanks for bringing us the new GNN-MLP SOTA. We will add this reference in the related section in a revised version. Although this method reported better performance against other GNN-MLP methods like GLNN and NOSMOG, it does not impact the novelty of our work. First, like other GNN-MLP methods, VQGraph fully relies on the target node features, which cannot solve the case that the label could rely on the features of neighbour nodes. Second, VQGraph trains the target GNN with the coded node features and essentially implicitly changes the target GNN models, which could bring better or worse performance. As shown in Fig. 7 from the VQGraph paper, it does not consistently have improved performance with better target GNN models, while from Table 2, our method is designed to approximate the target GNN and consistently improves with better target GNN models.
>
> __More experiments on heterophilic datasets.__ We agree that adding more experiments on heterophilic datasets will strengthen our work. We believe the current experiments are sufficient to demonstrate the advantages of SDGNN. We will apply those experiments in future work.
>
> __Asymptotic analysis.__ The asymptotic convergence rate in terms of the number of iterations is very challenging due to the non-convex nature of the loss function and the stochastic characteristics of the training algorithm. As for the candidate node generation algorithm, we will generate it once and use the same set of candidates throughout the optimization. Therefore, it is not the dominating factor. For the LARS solver, the per-iteration complexity will be $|\mathbb{B}|(|\mathbb{C}_z|^3+|\mathbb{C}_z|^2d)$, where $|\mathbb{B}|$ is the batch size, $|\mathbb{C}_z|$ is the size of the candidate set and $d$ is the embedding dimension. We will add the complexity analysis in a revised version.
>
> __Sensitivity Analysis.__ $\lambda_2$ is a common hyper-parameter for the strength of L2 regularizor. $\lambda_1$ controls the sparsity. In our implementation, since we are using LARS, there is no explicit $\lambda_1$, and the sparsity is controlled via maximum LARS iteration. The performance will generally increase if we enlarge the max LARS iterations, but the improvement will quickly saturate once the max LARS iterations reach some threshold. As for $K_1$ and $K_2$, they jointly determine the size of the candidate node set. We attached a training curve for the Pubmed/SAGE case with different settings of $K_1$ with $K_2=0$ (see Fig. 3 in \url{https://docs.google.com/document/d/e/2PACX-1vRNHjyltQZUQs_IM0Ota0Zvm0eUO2PRbg1i-i02igGq9DsnIB3EANvRcn8w41R4NRN-b1JfqNnT2Ctw/pub}). Note that $K_1$ is the dominating factor for the size of the candidate and varying different $K_2$ will yield some intermediate results. We can see that the loss will converge faster with a small candidate size, but they will converge to some sub-optimal point. With a proper size ($K_1=3$ in this case), SDGNN will robustly and efficiently converge to some decent solution. We will add more elaborations on the impact of hyper-parameters and sensitivity experiments in the revised version.
>
> __Training time/space cost.__ We will add a detailed analysis of the time and space cost in a revised version. The time cost is indeed no cheaper than to train SAGE. The design of stochastic training and candidate pruning accommodates large-scale graphs, which helped us successfully train the SDGNN for the OGBN-Products in about 20 hours. Compared to GNNs like SAGE, the extra space cost comes from storing the sparse weights, which is proportional to the number of nodes and the number of non-zero entries of those weights.

---

> > ### Comment · Reviewer_M5hg · 2025-01-12
> > **Thank you**
> >
> > **Intuition of SDGNN.**
> >
> > Thank you for the clearer explanation.
> >
> > **Importance of components of SDGNN**
> >
> > Thank you for more details on how settings of each component impacts downstream behavior. It makes sense that LARS iterations should saturate after some time, and that $K_1$ and $K_2$ can improve inference time. However, I cannot say very much about this without tables or experimental results. I look forward to these results in a revised version.
> >
> > **Significance of dynamic node features**
> >
> > Thank you for this explanation.
> >
> > **Compare with VQGraph**
> >
> > Can you explain more by what you mean by "VQGraph fully relies on the target node features, which cannot solve the case that the label could rely on the features of neighbor nodes."? Also, I understand that VQGraph is not guaranteed to perform at least as well as its target GNN. My main request is to at least see how SDGNN compares with VQGraph since both methods are aware of teh imoprtance of incorporating neighborhood features.
> >
> > **Heterophilic datasets**
> >
> > I look forward to these experiments.
> >
> > **Asymptotic analysis**
> >
> > Thank you for this analysis.
> >
> > **Sensitivity analysis**
> >
> > The Google Doc link seems to be broken. Otherwise, thank you for the added figure. I look forward to more elaborations in a revision.
> >
> > **Training time/space cost**
> >
> > Thank you for reporting this number of 20 hours. I look forward to a more detailed analysis in a revision.
> >
> > Overall, it is difficult to say I am satisfied without the studies and elaborations promised in a revision. Since TMLR allows updates to submissions, I would be more convinced of the rigor of this work once these studies are uploaded.

---

> > > ### Author Response · Authors · 2025-01-14
> > > **Thanks for the feedback**
> > >
> > > Thanks again for your prompt feedback.
> > >
> > > __Broken link__ Here is the updated link to the Google doc https://docs.google.com/document/d/e/2PACX-1vRNHjyltQZUQs_IM0Ota0Zvm0eUO2PRbg1i-i02igGq9DsnIB3EANvRcn8w41R4NRN-b1JfqNnT2Ctw/pub
> > >
> > > __VQgraph__ At the inference stage, all the GNN-MLP variants, including VQgraph, solely rely on that node's features to predict the node's label. From the angle of the input during inference, all the GNN-MLP variants cannot handle the features of neighbour nodes. Let's consider an extreme case that this would fail for sure. Given a graph, let the node feature be some random colour, and the label is set to be the colour which most first-hop neighbours have. We can see that the label of each node is independent of its own feature. GNN-MLP variants will fail since they cannot get critical inputs.  On the other hand, if we select one of the neighbours with the label colour in SDGNN, the model can capture this information.

---

### Review · Reviewer_bvcp · 2024-11-27

**Summary Of Contributions:**

The paper introduces SDGNN, a novel method for efficiently approximating GNN node representations.
The key innovation is combining learned feature transformations with sparse weight vectors to determine which transformed node features to aggregate, allowing efficient approximation of GNN embeddings.

Unlike prior MLP-based approximations, SDGNN can process neighbor features during inference while maintaining linear complexity.
It provides a principled way to trade off between approximation quality and inference efficiency through sparsity constraints, though with some limitations in inductive settings and non-linear feature interactions.

**Audience:**

Yes

**Claims And Evidence:**

Yes

**Requested Changes:**

**Requested Changes**

1. Include a theoretical/empirical analysis section examining SDGNN's expressiveness compared to standard GNNs. At minimum, identify conditions under which SDGNN can/cannot match/imporve GNN performance.

**Suggested Improvements**

1. Add experiments showing how performance degrades with increasing time gaps between training and testing in the spatio-temporal setting.
2. Include an analysis of memory requirements alongside the existing computational complexity discussion.
3. Provide a detailed discussion of strategies for handling distribution shift in online/streaming settings. This is essential since the method is aimed at online prediction scenarios where data distributions may evolve over time.
4. Add experiments on more diverse types of graph data beyond citation networks and traffic forecasting.
5. Provide guidelines for selecting key hyperparameters like K1 and K2 based on graph properties.

**Strengths And Weaknesses:**

**Strengths**

1. The work addresses an important practical problem of making GNN inference more efficient for online prediction while preserving model quality. The solution is well-motivated and technically sound.
2. The empirical validation is thorough and convincing, with extensive experiments across different datasets, model architectures, and application settings (node classification and spatio-temporal forecasting).
3. The method is flexible and can approximate various GNN architectures while maintaining linear complexity. The authors demonstrate this by successfully applying SDGNN to GraphSAGE, Geom-GCN, Exphormer and other architectures.
4. The authors provide clear comparisons with existing approaches and demonstrate meaningful improvements over state-of-the-art methods.

**Weaknesses**

1. The theoretical analysis is limited. The paper would benefit from formal analysis of SDGNN's expressiveness compared to standard GNNs and theoretical guarantees on approximation quality. I'm particularly interested in why the GNN's performances are improved after the decomposition. Is it possible to add some (heuristic) explanations? Such as visualise these embeddings?
2. The approach has limited ability to capture non-linear interactions between node features, which may be important for some applications. This fundamental limitation deserves more discussion.
3. The method cannot handle inductive settings where new nodes appear during inference. While the authors acknowledge this limitation, more discussion of potential solutions would be valuable.

---

> ### Author Response · Authors · 2025-01-04
> **Reply for Comments Part 1: the Requested Changes**
>
> We thank you for your constructive and inspiring feedback.
>
> __Expressiveness of SDGNN.__ We need to clarify that we aim to find a fundamental and model-agnostic solution to approximate the prediction of any target GNN model with efficient computation at the inference stage rather than inventing an efficient model with great performance. Therefore, the performance of SDGNN is designed to be similar to the target GNN models, which has been empirically demonstrated in Table 2 across extensive datasets and target GNN models. Other GNN-MLP works, such as NOSMOG, implicitly integrated additional modules to boost the performance of GNN (NOSMOG extend the distillation with augmented features), and that's why they could have potentially different performance than the target GNN. From Table 2, we can see that they will have better performance than a weak GNN target like SAGE (which is what they reported in their paper) but cannot achieve similar performance of stronger GNN models.
>
> Then, the question remains: under which condition can the SDGNN match the performance of a target GNN? This is really an interesting but challenging question, and we have not reached a complete answer for general cases. However, we have found some insights w.r.t. to a group of spectral GNN models. Inspired by the definitions in Sec. 2.2 of [1], we have found that the Filter-Most-Expressive GNNs can be well approximated via SDGNN with a theoretical guarantee. Specifically, it is defined as
>
> $\psi(\alpha(\mathbf{L})\varphi(\mathbf{X})),$
>
> where $\mathbf{X}\in\mathbb{R}^{|\mathcal{V}|\times D}$ is the node feature matrix with $|\mathcal{V}|$ nodes and $D$ feature dimension, $\varphi(\cdot)$ and $\psi(\cdot)$ denotes MLPs operating on the rows of the input matrix, $\alpha(\cdot)$ denotes arbitrary real value filter function and $\mathbf{L}$ is the normalized graph Laplacian. This form covers a wide range of spectral GNNs (c.f. Appendix A from [1]).
>
> Let's briefly analyze why SDGNN can cover the Filter-Most-Expressive GNNs. In SDGNN, if the downstream decoding function $f(\cdot)$ (in Sec. 6.1.5) is set to be $\psi(\cdot)$ and the feature transformation function $\phi(\cdot)$ is set to be $\varphi(\cdot)$, the $\mathbf{\theta}_z$ can be regarded as picking a few critical non-zero entries for the $z$th row of $\alpha(\mathbf{L})$ so that the result from SDGNN is as close as possible to the $z$th row of $\alpha(\mathbf{L})\varphi(\mathbf{X})$. A sufficient condition would be that $\|\alpha(\mathbf{L})- \mathbf{\Theta}\circ\alpha(\mathbf{L})\|_2$ is small where $\|\cdot\|_2$ denotes the spectral norm and $\circ$ denotes the element-wise product operation. [2] indicates that for any positive number $\epsilon>0$, there exists a $\mathbf{\Theta}$ with $O(|\mathcal{V}|/\epsilon^2)$ non-zero entries, such that $\|\alpha(\mathbf{L})- \mathbf{\Theta}\circ\alpha(\mathbf{L})\|_2\le\epsilon|\mathcal{V}|$ if $\alpha(\mathbf{L})$ is positive semidefinite. For general $\alpha(\mathbf{L})$, it satisfies that $\|\alpha(\mathbf{L})- \mathbf{\Theta}\circ\alpha(\mathbf{L})\|_2\le\epsilon~\text{max}(|\mathcal{V}|, \|\alpha(\mathbf{L})\|_1)$, where $\|\cdot\|_1$ denotes the nuclear norm.
>
> These facts indicate that SDGNN can well approximate the Filter-Most-Expressive GNNs, which cover a wide range of popular spectral GNNs. We will add this analysis in a revised version.
>
> __Performance gap with increasing time gaps.__ We have included extra results for this purpose in Fig. 1 and Fig. 2 in  https://docs.google.com/document/d/e/2PACX-1vRNHjyltQZUQs_IM0Ota0Zvm0eUO2PRbg1i-i02igGq9DsnIB3EANvRcn8w41R4NRN-b1JfqNnT2Ctw/pub . We plot the absolute error with standard deviation from 10 equal-sized chunks split by time. We can see that the performance will be worse when time gaps are larger from the training set.
>
> [1] Wang, Xiyuan, and Muhan Zhang. "How powerful are spectral graph neural networks." International conference on machine learning. PMLR, 2022.
>
> [2] Bhattacharjee, Rajarshi, et al. "Universal Matrix Sparsifiers and Fast Deterministic Algorithms for Linear Algebra." Innovations in Theoretical Computer Science (ITCS 2024) (2024).

---

> ### Author Response · Authors · 2025-01-04
> **Reply for Comments Part 2: the Requested Changes cont.**
>
> __Memory requirement.__ We will replace Table 2 with the following table with memory requirements. All method requires $O(|\mathcal{V}|D)$ to store the node features. Besides, SDGNN requires extra space of $O(|\mathcal{V}|\bar{d}L)$ to store the sparse weights, which have a similar size to the node features in the empirical results. $L$: number of the target GNN layers; $L^\prime$: number of a shallower GNN layers where $L^\prime<L$; $|\mathcal{V}|$: node number; $\bar{d}$: average node degree; $s$: sampling budget; $D$: node feature dimension.
>
> | Related Techniques         |   Inference complexity  |                Memory overhead                | Neighbour  feature-aware | Dynamic  features |
> |----------------------------|:-----------------------:|:---------------------------------------------:|:------------------------:|:-----------------:|
> | Neighbour Sampling         |         $O(s^L)$        | $O(\|\mathcal{V}\|\bar{d}+\|\mathcal{V}\|D))$ |      \CheckmarkBold      |   \CheckmarkBold  |
> | Embedding Compression      |      $O(\bar{d}^L)$     |  $O(\|\mathcal{V}\|\bar{d}+\|\mathcal{V}\|D)$ |      \CheckmarkBold      |   \CheckmarkBold  |
> | GNN Knowledge Distillation | $O(\bar{d}^{L^\prime})$ |  $O(\|\mathcal{V}\|\bar{d}+\|\mathcal{V}\|D)$ |      \CheckmarkBold      |   \CheckmarkBold  |
> | MLP Knowledge Distillation |          $O(1)$         |             $O(\|\mathcal{V}\|D)$             |       \XSolidBrush       |   \CheckmarkBold  |
> | Heuristic Decoupling       |          $O(1)$         |             $O(\|\mathcal{V}\|D)$             |      \CheckmarkBold      |    \XSolidBrush   |
> | SDGNN (ours)               |      $O(\bar{d}L)$      | $O(\|\mathcal{V}\|\bar{d}L+\|\mathcal{V}\|D)$ |      \CheckmarkBold      |   \CheckmarkBold  |
>
> __Handling the Distribution shift.__ A typical and practical solution is periodically retraining new models based on recently collected data. This can solve the distribution shift issue for both the target GNN model and SDGNN. If we trade the performance for less computation overhead, we can periodically and partially update SDGNN for better training efficiency. For example, we can keep the weight matrix $\mathbf{W}$ unchanged and only periodically perform a Phase $\mathbf{\Theta}$ update according to the latest data. A fundamental approach could incorporate online learning so that the trainable weights from the target GNN and SDGNN can be adapted as more data become available and the graph topology changes. However, developing such a method is involved due to the challenges of enforcing a sparse solution. We would leave this as a future work. We will add more details in a revised version.
>
> __More experiments on diverse datasets.__ We agree that adding more experiments will strengthen our work. We believe the current experiments are sufficient to demonstrate the advantages of SDGNN. Note that the OGBN-Products dataset is not from a citation network but is large-scale, well-accepted and challenging data from the Amazon product co-purchasing network. We want to leave the discussion for more diverse datasets in some future works.
>
> __Guideline to select $K_1$ and $K_2$.__ We don't have a general optimal method for selecting these parameters. The strategy we were adopting was to set them to make the size of the candidate nodes similar to or slightly larger than the size of receptive nodes from the target GNN models. Specifically, We first set $K_1$ according to this criterion with $K_2=0$. Then, we add more nodes by searching $K_2$ with different positive numbers.

---

> ### Author Response · Authors · 2025-01-04
> **Reply for Comments Part 3: the Weaknesses**
>
> __Theoretical analysis.__ Please refer to the response in the expressiveness of GNN.
>
> __Non-linear interactions between node features.__ From the analysis for expressiveness, SDGNN can well approximate the Filter-Most-Expressive GNNs, which cover lots of well-known GNNs. We will add more arguments for the non-linear feature interaction issue. For theoretical analysis on a more general case, we will leave it to future work.
>
> __Handling new nodes in the inductive setting.__ I should admit that I don't have a principled approach to solving this issue with the current algorithm. A simple engineering strategy to handle the novel nodes in real-time could be collecting the $\mathbf{\theta}$ for all/sampled $1$-hop neighbour nodes and summing them up as the proxy for the new nodes. Moreover, for the situation where we are allowed to do some preprocessing for the novel nodes, we could use the inductive GNN to infer the GNN embedding, compute $\mathbf{\theta}$ with Phase $\mathbf{\theta}$ for that node and cache it. Then, the actual online inference stage will be the same as the other nodes that are seen during the training of SDGNN. We will add more elaboration in a revised version.

---

### Review · Reviewer_Vd3Z · 2024-12-19

**Summary Of Contributions:**

The paper presents an approach called SDGNN that reduces inference time by approximating node representations through a weighted sum of features from a carefully selected subset of nodes, achieving linear complexity with respect to the average node degree and the number of layers.

**Audience:**

Yes

**Claims And Evidence:**

Yes

**Requested Changes:**

See Weaknesses

**Strengths And Weaknesses:**

Strengths:
- The paper is well-structured and easy to follow, with clear explanations of the proposed method and its components.
- The experimental results are comprehensive, demonstrating the effectiveness of SDGNN across various datasets and tasks.
- The approach is innovative in its use of sparse decomposition to maintain model accuracy while significantly reducing inference time.

Weaknesses:
- How does the method handle the introduction of new nodes in an inductive learning setting, given that it requires pre-computed embeddings for all nodes?
- Is there a risk of losing important non-linear interactions between node features due to the linear combination used in the sparse decomposition?
- How does the training time and computational complexity scale with very large graphs?
- How does the method ensure that the sparse weight vectors θz remain non-negative during the optimization process, as indicated by the constraint in equation (5)?
- In the context of narrowing the candidate nodes, how does the heuristic approach for selecting the candidate set impact the model's ability to capture important node features?

---

> ### Author Response · Authors · 2025-01-04
> **Reply for Comments Part 1: the Weaknesses**
>
> Thank you for your constructive and inspiring feedback on our work.
>
> __Handling new nodes in the inductive setting.__ We need to clarify that our method pre-computes the optimal and sparse aggregation weights over a few nodes rather than the ``embeddings'' for all nodes. We admit that our method was not designed to handle the new nodes in a principled way. In the appendix, we pointed it out in Sec. D.1 that some native engineering approaches can process the new nodes. A simple strategy could be collecting the $\mathbf{\theta}$ for all/sampled $1$-hop neighbour nodes and summing them up as the proxy for the new nodes. Moreover, for the situation where we are allowed to do some preprocessing for the novel nodes, we could use the inductive GNN to infer the GNN embedding, compute $\mathbf{\theta}$ with Phase $\mathbf{\theta}$ for that node and cache it. Then, the actual online inference stage will be the same as the other nodes that are seen during the training of SDGNN. We will add more elaboration in a revised version.
>
> __Non-linear interactions.__ We don't have the complete answer for the general GNNs, but we have found some theoretical guarantees for a group of GNN models. Inspired by the definitions in Sec. 2.2 of [1], we have found that SDGNN has sufficient expressive power to approximate the Filter-Most-Expressive GNNs without the risk of losing important non-linear interactions between node features. Specifically, it is defined as
>
>  $\psi(\alpha(\mathbf{L})\varphi(\mathbf{X})),$
>
> where $\mathbf{X}\in\mathbb{R}^{|\mathcal{V}|\times D}$ is the node feature matrix with $|\mathcal{V}|$ nodes and $D$ feature dimension, $\varphi(\cdot)$ and $\psi(\cdot)$ denotes MLPs operating on the rows of the input matrix, $\alpha(\cdot)$ denotes arbitrary real value filter function and $\mathbf{L}$ is the normalized graph Laplacian. This form covers a wide range of spectral GNNs (c.f. Appendix A from [1]).
>
> Let's briefly analyze why SDGNN can cover the Filter-Most-Expressive GNNs. In SDGNN, if the downstream decoding function $f(\cdot)$ (in Sec. 6.1.5) is set to be $\psi(\cdot)$ and the feature transformation function $\phi(\cdot)$ is set to be $\varphi(\cdot)$, the $\mathbf{\theta}_z$ can be regarded as picking a few critical non-zero entries for the $z$th row of $\alpha(\mathbf{L})$ so that the result from SDGNN is as close as possible to the $z$th row of $\alpha(\mathbf{L})\varphi(\mathbf{X})$. A sufficient condition would be that $\|\alpha(\mathbf{L})- \mathbf{\Theta}\circ\alpha(\mathbf{L})\|_2$ is small where $\|\cdot\|_2$ denotes the spectral norm and $\circ$ denotes the element-wise product operation. [2] indicates that for any positive number $\epsilon>0$, there exists a $\mathbf{\Theta}$ with $O(|\mathcal{V}|/\epsilon^2)$ non-zero entries, such that $\|\alpha(\mathbf{L})- \mathbf{\Theta}\circ\alpha(\mathbf{L})\|_2\le\epsilon|\mathcal{V}|$ if $\alpha(\mathbf{L})$ is positive semidefinite. For general $\alpha(\mathbf{L})$, it satisfies that $\|\alpha(\mathbf{L})- \mathbf{\Theta}\circ\alpha(\mathbf{L})\|_2\le\epsilon~\text{max}(|\mathcal{V}|, \|\alpha(\mathbf{L})\|_1)$, where $\|\cdot\|_1$ denotes the nuclear norm.
>
> These facts indicate that SDGNN can well approximate the Filter-Most-Expressive GNNs without the issue of losing important non-linear interactions. For more general cases, we will leave the theoretical analysis in future work. We will add this analysis in a revised version.
>
> [1] Wang, Xiyuan, and Muhan Zhang. "How powerful are spectral graph neural networks." International conference on machine learning. PMLR, 2022.
>
> [2] Bhattacharjee, Rajarshi, et al. "Universal Matrix Sparsifiers and Fast Deterministic Algorithms for Linear Algebra." Innovations in Theoretical Computer Science (ITCS 2024) (2024).

---

> ### Author Response · Authors · 2025-01-04
> **Reply for Comments Part 2: the Weaknesses conts.**
>
> __Scalability.__ The structure of SDGNN guarantees low inference computation complexity, while the mini-batch training and candidate narrowing strategy makes the training of SDGNN scale to large graphs.
>
> The inference computation complexity is proportional to the number of receptive nodes, which is the number of non-zero entries in the sparse weight vector in SDGNN. The sparsity is guaranteed via the optimization algorithm.
>
> For the training time, the mini-batch sampling and narrowing of the candidate node sets make the per-iteration complexity scalable to large graphs. Specifically, mini-batch sampling makes each iteration's cost proportional to the batch size rather than the size of the graph. Within the LARS solver, the candidate selection strategy further reduces the complexity into $(|\mathbb{C}_z|^3+|\mathbb{C}_z|^2d)$ for each node, where $|\mathbb{C}|$ is the size of the candidate nodes and $d$ is the embedding size, which is independent of the graph size. Along with the Phase $\mathbf{W}$, the per-iteration complexity is $O(|\mathbb{B}|(k(Dd+L^\prime d^2) + |\mathbb{C}_z|^3+|\mathbb{C}_z|^2d))$, where $L^\prime$ is the number of layers in the feature transformation function and $k$ is the SGD iterations for Phase $\mathbf{W}$. We can see that per-iteration complexity is independent of the size of the graph.
>
> We have also demonstrated the scalability with the OGBN-products dataset, which is a well-known large-scale dataset with 2.4 million nodes. The training takes about 20 hours, and Figure 2 depicts that the inference complexity is much lower than the target GNN models.
>
> __How to enforce the non-negative condition in (5).__ The Least Angle Regression algorithm implemented in the Scikit-learn package has a built-in option of it. We will provide some details about this in a revised version.
>
> __Impact of narrowing the candidate nodes.__ We don't have a complete theoretical guarantee for this procedure, but the rule of thumb is to set them to make the size of the candidate nodes similar to or slightly larger than the size of receptive nodes from the target GNN models. We attached a training curve for the Pubmed/SAGE case with different settings of $K_1$ with $K_2=0$ (see Fig. 3 in  https://docs.google.com/document/d/e/2PACX-1vRNHjyltQZUQs_IM0Ota0Zvm0eUO2PRbg1i-i02igGq9DsnIB3EANvRcn8w41R4NRN-b1JfqNnT2Ctw/pub). Note that $K_1$ is the dominating factor for the size of the candidate and varying different $K_2$ will yield some intermediate results. We can see that the overly shrunk candidate set will result in sub-optimal performance. In practice, we need some hyper-parameter search to strike a balance between the training complexity and the overall performance. We will add more elaborations for this issue in a revised version.

---

### Decision · Action_Editor_HPEj · 2025-02-14

**Recommendation:** Accept with minor revision

**Comment:**

The paper presents SDGNN, a novel approach for efficient GNN inference through sparse decomposition. All three reviewers recommend acceptance (1 Accept, 2 Leaning Accept), highlighting:

- Major Strengths:

1. Novel and well-motivated approach combining learned feature transformations with sparse weight vectors
2. Strong theoretical foundation with guarantees for approximating Filter-Most-Expressive GNNs
3. Comprehensive empirical validation across diverse datasets
4. Clear practical value with significant efficiency improvements
5. Thorough responses to reviewer concerns with additional analyses and experiments

- Key Improvements Made:

1. Added theoretical analysis of expressiveness
2. Provided memory requirement analysis
3. Included temporal performance degradation studies
4. Added sensitivity analyses for hyperparameters
5. Clarified handling of dynamic node features

- Remaining Considerations:

1. Addition of VQGraph comparison recommended
2. Further evaluation on heterophilic datasets suggested
3. More detailed training cost analysis would be valuable

The authors have been highly responsive to reviewer feedback, providing detailed theoretical analyses, additional experiments, and clarifications. The work makes both theoretical and practical contributions while maintaining strong empirical validation.

- Recommendation: Accept with encouragement to address remaining minor points in final version, particularly:

Expanding heterophilic dataset evaluation
Including comprehensive training cost analysis

The paper represents a significant contribution to efficient GNN inference with both theoretical and practical value.

**Audience:**

Yes, this work would be of strong interest to TMLR's audience for several reasons:

1. Addresses practical challenge of efficient GNN inference
2. Provides theoretical insights about GNN approximation
3. Introduces novel method applicable to various GNN architectures
4. Offers practical solutions for resource-constrained settings
5 .Relevant to both theoretical researchers and practitioners

**Claims And Evidence:**

Claims And Evidence:
The paper's claims about SDGNN are well-supported by evidence:

- Strengths:

1. Comprehensive experimental validation across various datasets and tasks
2. Clear empirical demonstration of improved inference efficiency
3. Thorough theoretical analysis showing SDGNN can approximate Filter-Most-Expressive GNNs
4. Detailed ablation studies and sensitivity analyses added in response to reviewer requests
5. Strong performance comparisons against existing methods

- Limitations:

1. Initial lack of analysis on memory requirements (addressed in revision)
2. Some benchmarks missing in original submission (VQGraph comparison requested)
3. Limited evaluation on heterophilic datasets
4. Training time costs analysis could be more detailed